# MergeTok: Unified Continuous and Discrete Visual Tokenization via Token Merging

## Abstract

Most existing visual tokenizers for image generation are bifurcated into two families with complementary limitations: continuous VAEs offer high-fidelity reconstruction but suffer from dense, entangled latents that are poorly suited for semantic control, whereas discrete VQ-based models enable autoregressive generation yet struggle with gradient sparsity, unstable training, and codebook collapse. In this work, we introduce MergeTok, a unified tokenizer that jointly optimizes continuous (VAE) and discrete (VQ) tokenizers within a shared encoder-decoder architecture, leveraging token merging techniques as a semantic bridge. By clustering similar tokens during encoding, MergeTok establishes a shared structural prior that provides dual supervision signals: (i) it imposes merged-token semantic alignment in the VAE branch, regularizing its latent space toward disentangled, semantic-aware representations; (ii) it derives group-wise constraints, promoting intra-group diversity and inter-group exclusivity that stabilize VQ training. MergeTok shows competitive reconstruction and generation performance on ImageNet-256, with substantially lower rFID than strong VAE-only and VQ-only models under matched token budgets, while producing compact, high-fidelity sequences compatible with both autoregressive and diffusion generators. This shows that a single architecture can simultaneously endow visual tokenizers with robust semantic organization and generator-friendly discreteness.

## 1. Introduction

Image generation has achieved remarkable success with auto-regressive (Esser et al., 2021; Zhang et al., 2025) and diffusion-based approaches (Peebles & Xie, 2023). This relies heavily on the underlying visual tokenizer, typically an

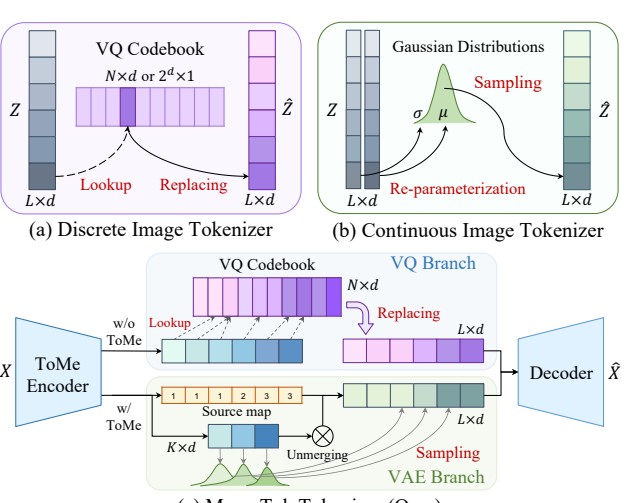

*Figure 1.* **(a) Discrete (VQ):** Discretizes features through nearest-neighbor codebook lookup, often suffering from gradient sparsity. **(b) Continuous (VAE):** Maps features to Gaussian latent and then sampled via re-parameterization, prioritizing reconstruction. **(c) MergeTok.** A unified architecture that bridges both manners via token merging (ToMe) (Bolya et al., 2023). A shared ToMe encoder aggregates semantically similar tokens with a source map $S$ that (i) regularizes VAE by merged token alignment and (ii) guides VQ with group-aware quantization, allowing both branches to be jointly trained for discrete and continuous tokenization.

auto-encoder architecture that maps an image to a sequence of feature representations and then applies a *latent-space constraint* that encodes these features, either as a continuous distribution or discrete codes. Previous work falls into two groups based on latent constraints. **VQ-based** tokenizers like VQ-VAE (van den Oord et al., 2017) quantize encoder outputs to discrete codebook indices, enabling sequence modeling via categorical distributions. In contrast, **VAE-based** methods map features to a continuous probabilistic distribution, prioritizing reconstruction fidelity and gradient flow (Kingma, 2013; Yao & Wang, 2025). It is commonly conjectured that improving the encoder and latent modules of tokenizers is crucial as their capacity often governs the efficiency and semantic controllability of downstream generative models (Kim et al., 2025; Li et al., 2025).

Despite the notable progress, both paradigms face structural, complementary limitations. Continuous VAEs suffer from a trade-off between semantic disentanglement and latent

[1]Anonymous Institution, Anonymous City, Anonymous Region, Anonymous Country. Correspondence to: Anonymous Author <anon.email@domain.com>.

Preliminary work. Under review by the International Conference on Machine Learning (ICML). Do not distribute.

compactness: to preserve fine-grained details under high compression, encoders learn dense, rich latents that entangle factors of variation, while enforcing disentanglement (*e.g.*, $\beta$-VAE (Kingma, 2013)) degrades the reconstruction quality, making it difficult to obtain compact codes that are both high-fidelity and structured for generation. In contrast, discrete VQ tokenizers struggle with optimization pathologies: non-differentiable assignment and straight-through estimation yield sparse, delayed gradients, leading to codebook collapse, unstable and suboptimal encoder–decoder training. This reveals a persistent tension among fidelity, semantic structure, token-budget efficiency, and training stability.

To address these issues, we propose MergeTok, a unified, multi-task tokenizer that jointly optimizes a VAE branch and a VQ branch as shown in Fig. 1. Token merging clusters content-similar tokens at encoding time; we exploit this in two ways: (i) we impose merged-token alignment to regularize the VAE latent space toward disentangled, semantics-aware representations; and (ii) we use the induced source matrix to drive group-aware quantization, encouraging in-group code diversity and between-group exclusivity, stabilizing VQ training. A shared encoder/decoder is trained end-to-end under a single objective, with the VAE path maintaining continuous gradients that mitigate VQ's update sparsity, while VQ's global clustering injects semantic structure back into the latent space. Experiments on ImageNet-256 generation and reconstruction benchmarks show remarkable rFID and perceptual quality, with compact sequences that are friendly to both AR and diffusion generators, consistently outperforming strong VAE-only and VQ-only baselines. Our contribution can be summarized as:

- **A unified dual-branch system.** We propose *MergeTok*, a tokenizer that jointly optimizes *continuous* and *discrete* latents under a single objective. By treating token merging as a semantic bridge, we enable an encoder-decoder architecture where the continuous stability of VAE and discrete structure of VQ regularize each other.

- **Merge-aware constraints.** We introduce two simple yet effective constraints that leverage the merging process: (i) *merged-token alignment* to enforce semantic disentanglement in the continuous VAE space and (ii) *group-aware quantization* to stabilize the VQ training.

- **Granularity-aware merge ratio sampling.** A discrete merge-ratio sampling is employed during training. This exposes the model to multiple token granularities, enabling multi-task tokenization with improved fidelity and efficiency across reconstruction and generation.

## 2. Related Work

**Visual Tokenizers for Image Generation.** Modern visual tokenizers map images to discrete or continuous sequences for AR-based generation. On the discrete side, recent methods mitigate gradient sparsity and codebook collapse by improving differentiability and scaling: IBQ propagates gradients over the full code distributions to boost code utilization (Shi et al., 2024); LFQ replaces vector lookups with binary indices for extreme vocabulary sizes (Yu et al., 2023a); FQ factorizes a large codebook into coordinated sub-codebooks with disentanglement and guidance; and SoftVQ-VAE adopts the categorical posteriors to attain high compression rate (Chen et al., 2025b).

*Continuous tokenizers instead progress along two axes: high compression and semantic organization.* DC-AE (Chen et al., 2025c) combines residualized latents with staged training to sustain quality at extreme spatial downsampling and to accelerate latent diffusion; (Chen et al., 2025a) shows that masked autoencoding can induce a more discriminative latent with far fewer tokens and substantially faster training; and representation alignment techniques (Yu et al., 2025) and its end-to-end variant) align latent features to powerful visual encoders, stabilizing joint optimization with diffusion and improving both convergence and generative quality. Collectively, these advances expand capacity, restore gradient flow, and impose semantic structure, enabling tokenizers that are both reconstruction-faithful and modeling-friendly. Recent efforts also explore joint VAE–VQ optimization. Some aim to improve the tokenizer itself, such as (Wang et al., 2025) and (Chen et al., 2025d) by coupling continuous and discrete objectives to stabilize codebooks and enrich semantics. Others use diffusion to strengthen autoregressive decoders (Li et al., 2024; Ren et al., 2024; Huang et al., 2025; Yang et al., 2025; Liu et al., 2025). Note that our proposed MergeTok differs by treating the visual tokenizer as a unified system and optimizing both continuous and discrete paths end-to-end via lightweight token merging techniques.

**Unified Tokenizers for Multimodal Learning.** UniTok (Ma et al., 2025) attributes the apparent reconstruction–semantics tension to limited discrete capacity and alleviates it via multi-codebook quantization and wider embeddings, achieving low rFID and strong zero-shot accuracy while integrating cleanly with MLLMs for native generation. AToken (Lu et al., 2025) extends unification to images, video, and 3D with a Transformer and 4D positional encoding, using adversarial-free perceptual objectives and a curriculum to support cross-modal generation within a single token space. SPAE (Yu et al., 2023b) aligns vision and language by translating images into multi-scale tokens consumable by frozen LLMs, enabling both understanding and synthesis; likewise, improved tokens in AR frameworks such as (Yu et al., 2023c) allow GPT-style generators to rival diffusion on standard benchmarks. Together, capacity scaling, multi-codebook design, multi-scale tokenization, and language alignment emerge as key levers for a single tokenizer that supports diverse tasks and modalities.

**Token Compression.** A critical bottleneck for generative models, particularly autoregressive ones, is the computational complexity of processing long token sequences. This has motivated a distinct body of work on token compression. Adaptive-length tokenization (Yu et al., 2024b) recurrently distills 2D patches into compact 1D sequences, allocating more tokens to complex content and fewer to simple scenes, thereby matching capacity to entropy and reducing unnecessary decoding steps. Fixed aggressive compression via improved tokenizers (Chen et al., 2025c;b) achieves tens of tokens per image, yielding order-of-magnitude speedups with competitive fidelity. At inference, token-merging methods (Bolya et al., 2023) dynamically fuse redundant tokens inside generative backbones to cut compute and memory with negligible perceptual loss; training-time designs (Li et al., 2025) integrate merging into the encoder to form short, quantized sequences while retaining mechanisms to recover fine detail at decode time. Together, adaptive token counts and token merging reduce step count and memory pressure, often improving overall quality by limiting error propagation, and make multimodal generation more tractable.

## 3. Methods

### 3.1. Problem and Motivation

As aforementioned, existing visual tokenizer research faces the following challenges:

**Challenge 1: Limitations of Continuous VAE-based Tokenizers.** VAE-based tokenizers rely on a continuous latent space, which often lacks intrinsic structure. Without additional constraints, the latent encoding becomes overly dense and entangled, mixing different semantic factors across the latent components. This results in a lack of disentanglement, restricting controlled conditional generation and making it difficult to achieve factorized representations.

**Challenge 2: Limitations of Discrete VQ Tokenizers.** VQ-based tokenizers face optimization challenges due to the non-differentiability of the quantization process. This requires approximating the gradient, which leads to sparse gradients and suboptimal updates to the codebook. As such, most codebook entries receive little gradient, risking codebook collapse and limiting the representation efficiency.

In the following sections, we explain our specific designs with MergeTok, which addresses these challenges by combining the strengths of both VAE and VQ tokenizers while mitigating their individual limitations.

### 3.2. MergeTok Framework

In this section, we introduce a two-branch MergeTok framework that jointly addresses the challenges of both continuous

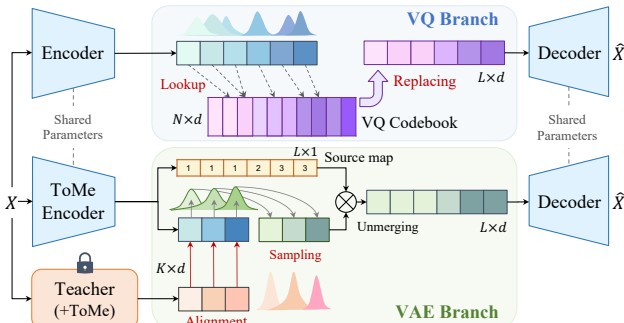

*Figure 2.* **Overall Framework of MergeTok.** We propose a dual-branch architecture that jointly optimizes continuous and discrete representations with shared encoder and decoder. **(i) VAE Branch (Bottom)** applies ToMe (Bolya et al., 2023) to extract dense semantic tokens, which are aligned with a teacher model (also equipped with ToMe). The resulting source map is then employed to unmerge the groups back to the full lattice for reconstruction. **(ii) VQ Branch (Top)** inherits this source map to induce group-aware clustering, enforcing intra-group diversity and inter-group exclusivity constraints that stabilize the training of the discrete codebook.

and discrete tokenizers as shown in Fig. 2. Given an input image $X \in \mathbb{R}^{H \times W \times 3}$, we encode it through a shared CNN encoder $\mathcal{E}_c(\cdot)$, resulting in a feature map $Z \in \mathbb{R}^{\frac{H}{f} \times \frac{W}{f} \times D}$, where $f$ is the downsampling factor, and $D$ is the channel dimension. The feature map $Z$ is reshaped into a token sequence of length $L = \frac{H}{f} \cdot \frac{W}{f}$, yielding $Z_L \in \mathbb{R}^{L \times D}$, i.e.,

$$Z_L = \mathcal{E}_c(X). \tag{1}$$

The resulting token sequence $Z_L$ serves as the input to both the VAE and VQ branches.

**VAE Branch.** The token sequence $Z_L$ is fed into an attention-based encoder $\mathcal{E}_a(\cdot; r)$ equipped with token-merging modules for further feature extraction, where $r \in (0, 1]$ controls the token keep ratio over $N$ layers. This produces a condensed representation $Z_K^{(\mathrm{vae})} \in \mathbb{R}^{K \times D}$ with $K = \lfloor \kappa L \rfloor$ (e.g., $\kappa = r^N$), together with a binary source map $S$ that preserves the original relative positional ancestry of the $K$ merged tokens by recording the assignment from the $L$ pre-merge tokens to the $K$ merged tokens:

$$Z_K^{(\mathrm{vae})}, \ S \ = \ \mathcal{E}_a(Z_L; r, N). \tag{2}$$

For reconstruction, we employ a hybrid VAE decoder $\mathcal{D}^{(\mathrm{vae})}(\cdot)$ that jointly takes the merged tokens and the source map to recover pixel-space details, yielding:

$$\hat{X}^{(\mathrm{vae})} \ = \ \mathcal{D}^{(\mathrm{vae})}\left(Z_K^{(\mathrm{vae})}, S\right). \tag{3}$$

We defer the implementation details to Sec. 3.3.

**VQ Branch.** The VQ branch uses the same attention encoder $\mathcal{E}_a(\cdot; r)$ but sets $r = 0$ to disable token merging,

| Original | MergeTok (VAE) w/ ToMe | MergeTok (VQ) w/o ToMe |
| --- | --- | --- |

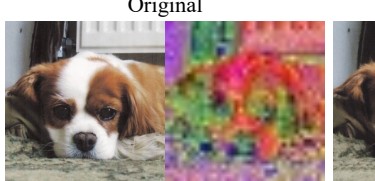 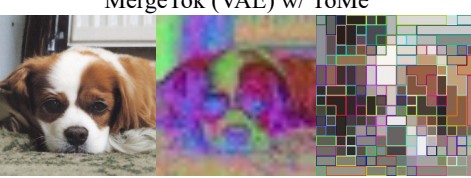 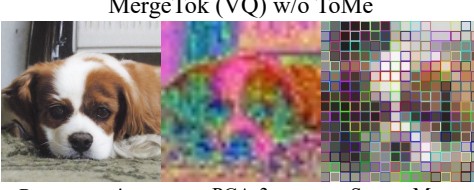

| Raw Image | PCA-3 | Reconstruction | PCA-3 | Source Map | Reconstruction | PCA-3 | Source Map |

*Figure 3.* **Semantic Condensing Effects.** Following DINOv2 (Oquab et al., 2024) and GigaTok (Xiong et al., 2025), we visualize the principal components (PCA-3) of the raw (or reconstruction) images and the resulting ToMe source maps (Bolya et al., 2023) to demonstrate how MergeTok organizes visual information. Holistically, the VAE branch is constrained by token-wise aggregation, enabling semantic separability comparable to discrete models. The VQ branch without ToMe possesses inherent clustering due to quantization.

preserving the full-length sequence:

$$Z_L^{(\text{vq})} \;=\; \mathcal{E}_a(Z_L;\, r{=}0) \in \mathbb{R}^{L \times D}. \qquad (4)$$

We quantize each token $z_t \in Z_L^{(\text{vq})}$ with a codebook $\mathcal{C} = \{c_j\}_{j=1}^n$ via:

$$\tilde{z}_t \;=\; \mathcal{Q}(z_t) = c_{i^*}, \;\; i^* \;=\; \arg\min_{j\in\{1,\dots,n\}} \|z_t - c_j\|_2, \quad (5)$$

yielding $\tilde{Z}_L^{(\text{vq})} = \{\tilde{z}_t\}_{t=1}^L$. During quantization, we further leverage the source map $S$ produced by the VAE branch to impose group-aware guidance on code assignments and improve codebook learning. Finally, a hybrid VQ decoder reconstructs the image:

$$\hat{X}^{(\text{vq})} \;=\; \mathcal{D}^{(\text{vq})}\!\left(\tilde{Z}_L^{(\text{vq})}\right). \qquad (6)$$

We defer the computation details to Sec. 3.4.

### 3.3. Semantic Enhancement in VAE Branch

To address Challenge 1 in Sec. 3.1, we adopt two strategies in the VAE branch: (1) *token merging for semantic abstraction* and (2) *matched-granularity semantic alignment*.

**Token Merging for Semantic Abstraction.** We apply the token merging algorithm in ToMe as the fusion module in our attention encoder $\mathcal{E}_a(\cdot)$. Concretely, $\mathcal{E}_a(\cdot)$ compresses the input sequence by merging the most similar tokens at each layer, controlled by a prescribed per-layer keep ratio $r \in (0, 1]$ over $N$ merging layers. Denoting the effective keep ratio by $\kappa = r^N$ (or more generally $\kappa = \prod_{\ell=1}^{N} r_\ell$ for layer-wise ratios $\{r_\ell\}$), the encoder outputs a condensed $K$-token representation $Z_K^{(\text{vae})} \in \mathbb{R}^{K \times D}$ with $K = \lfloor \kappa L \rfloor$, together with a source map $S \in \{1, \dots, K\}^{1 \times L}$ that records the assignment from original to merged tokens: the $i$-th entry $S[i]$ specifies the merged index, $S[i] = k$ means the original token $i$ is merged into the $k$-th token in $Z_K^{(\text{vae})}$. For reconstruction, we convert the source map $S$ into a one-hot assignment matrix $A \in \{0, 1\}^{L \times K}$ as:

$$A_{i,k} \;=\; \mathbf{1}[S[i] = k], \qquad A \in \{0, 1\}^{L \times K}. \qquad (7)$$

We then restore the original token layout via:

$$\tilde{Z}_L \;=\; A\, Z_K^{(\text{vae})} \in \mathbb{R}^{L \times D}. \qquad (8)$$

The recovered sequence $\tilde{Z}_L$ is decoded into the pixel-space $\hat{X}^{(\text{vae})}$ using a hybrid decoder as Eq. 3 . A more detailed computational example is provided in the Appendix A.

As such, the VAE branch discovers sample-wise semantic clusters at encoding time, decoupling high-frequency details and thereby preventing an overly dense latent space.

**Semantic Alignment at Matched Granularity.** To regularize the VAE latent space, we align merged tokens with a frozen DINO-style teacher $\mathcal{T}$ configured with the same merging schedule $(r, N)$. The teacher produces $K$ semantic features $F_K^{(\text{tea})} \in \mathbb{R}^{K \times D_t}$, and we project the merged latents $Z_K^{(\text{vae})} \in \mathbb{R}^{K \times D_t}$ via an alignment head $\mathcal{H}_{\text{ali}}$:

$$F_K^{(\text{tea})} = \mathcal{T}(X;\, r, N), \qquad \bar{Z}_K = \mathcal{H}_{\text{ali}}\!\left(Z_K^{(\text{vae})}\right). \quad (9)$$

We then compute the alignment loss $\mathcal{L}_{\text{align}}$ between the projected student features $\bar{Z}_K$ and the teacher features $F_K^{(\text{tea})}$ using a similarity metric as cosine distance:

$$\mathcal{L}_{\text{align}} = \frac{1}{K}\sum_{k=1}^{K} \left\| \bar{Z}_K[k] - F_K^{(\text{tea})}[k] \right\|_2^2, \qquad (10)$$

where $\bar{Z}_K[k]$ and $F_K^{(\text{tea})}[k]$ are the $k$-th merged token from the student and teacher, respectively.

This alignment loss encourages VAE latent tokens to capture meaningful structure aligned with a teacher model. Rather than aligning only the [CLS] token, which overlook fine-grained semantics, or all patch tokens, which can be overly rigid, we align at the semantic level using merged tokens. This facilitates focus on global semantics while reducing sensitivity to irrelevant details, leading to more coherent latent representations for generation as shwon in Fig. 3.

### 3.4. Improving VQ with VAE-Derived Group Priors

To address the Challenge 2 in Sec. 3.1, the VAE branch improves VQ by (1) *providing continuous gradients for*

*optimization*, and (2) *offering a clustering prior through token merging, where the source map $S$ guides quantization.*

**Continuous Gradient from VAE Enhances Encoder Optimization.** To address the non-differentiability of the quantization operation in Eq. 5, the VQ branch adopts the straight-through estimator to approximate gradients during backpropagation:

$$
\frac{\partial \mathcal{L}}{\partial \mathcal{E}} = \frac{\partial \mathcal{L}}{\partial \hat{X}^{(\mathrm{vq})}} \cdot \frac{\partial \hat{X}^{(\mathrm{vq})}}{\partial \tilde{Z}_L^{(\mathrm{vq})}} \cdot \frac{\partial \tilde{Z}_L^{(\mathrm{vq})}}{\partial Z_L} \cdot \frac{\partial Z_L}{\partial \mathcal{E}}
$$
$$
\approx \frac{\partial \mathcal{L}}{\partial \hat{X}^{(\mathrm{vq})}} \cdot \frac{\partial \hat{X}^{(\mathrm{vq})}}{\partial \tilde{Z}_L^{(\mathrm{vq})}} \cdot \frac{\partial \tilde{Z}_L^{(\mathrm{vq})}}{\partial \mathcal{E}}. \quad (11)
$$

While effective, this approximation leads to training instability. The VAE branch mitigates this by providing continuous gradients to the encoder, facilitating faster convergence.

**Clustering Prior from VAE Enhances VQ Quantization.** As in Eq. 4, the VQ branch produces the full-length latents $Z_L^{(\mathrm{vq})}$. The quantization is optimized with the standard VQ objective in Eq. 5, including the commitment loss

$$
\mathcal{L}_{\mathrm{com}} = \beta \left\| \mathrm{sg}\left[ Z_q^{(\mathrm{vq})} \right] - Z_L^{(\mathrm{vq})} \right\|_2^2, \quad (12)
$$

where $Z_q^{(\mathrm{vq})}$ denotes the quantized latents, $\mathrm{sg}[\cdot]$ is the stop-gradient operator, and $\beta$ is the weight. Building on this, we further refine VQ quantization using the VAE-derived source map $S$, which captures sample-wise token grouping. This enables two group-aware regularizers: an *intra-group diversity loss* $\mathcal{L}_{\mathrm{div}}$ that encourages diverse code usage within each group, and an *inter-group consistency loss* $\mathcal{L}_{\mathrm{cons}}$ that discourages code sharing across different groups.

As in Eq. 2 and Eq. 7, the VAE branch provides a source map $S$ and its one-hot variant $A \in \{0,1\}^{L \times K}$, where $A_{i,g} = 1$ indicates that the $i$-th original token belongs to group $g$. Let $q_i \in \{1, \ldots, n\}$ be the hard code index assigned to the $i$-th VQ token (codebook size $n$). As such, the group size is:

$$
N_g = \sum_{i=1}^{L} A_{i,g}. \quad (13)
$$

We then summarize code usage *within each group* by a categorical distribution $p_g \in \mathbb{R}^n$, whose $k$-th entry counts the fraction of tokens in group $g$ assigned to code $k$:

$$
p_{g,k} = \frac{1}{N_g} \sum_{i=1}^{L} A_{i,g} \mathbf{1}[q_i = k], \qquad k \in \{1, \ldots, n\}. \quad (14)
$$

Since $\sum_{k=1}^{n} \mathbf{1}[q_i = k] = 1$ for each token $i$, we have $\sum_{k=1}^{n} p_{g,k} = 1$, $p_g$ is a valid probability distribution over code indices for group $g$.

*Intra-group Diversity Loss.* To prevent a semantic group from collapsing to a single code, we maximize the entropy of $p_g$. Equivalently, we minimize the negative entropy:

$$
\mathcal{L}_{\mathrm{div}} = -\sum_{g=1}^{K} H(p_g) = \sum_{g=1}^{K} \sum_{k=1}^{n} p_{g,k} \log p_{g,k}. \quad (15)
$$

where $H(p_g) = -\sum_k p_{g,k} \log p_{g,k}$ is maximized when codes are used more evenly within group $g$, thus encouraging *diverse-within* code assignments and improving codebook utilization.

*Inter-group Consistency Loss.* While $\mathcal{L}_{\mathrm{div}}$ encourages diversity within each group, we additionally enforce *separation between groups* by discouraging different groups from using the same codes. We measure the overlap between two groups $g$ and $h$ by the dot product of their code-usage distributions:

$$
\langle p_g, p_h \rangle = \sum_{k=1}^{n} p_{g,k} \, p_{h,k}. \quad (16)
$$

This quantity is large when both groups place mass on the same set of codes, and small when they use disjoint code subsets. Aggregating this overlap over all distinct group pairs yields:

$$
\mathcal{L}_{\mathrm{cons}} = \sum_{\substack{g,h=1 \\ g \neq h}}^{K} \sum_{k=1}^{n} p_{g,k} \, p_{h,k} = \sum_{\substack{g,h=1 \\ g \neq h}}^{K} \langle p_g, p_h \rangle. \quad (17)
$$

Minimizing $\mathcal{L}_{\mathrm{cons}}$ therefore drives different groups to specialize on different code subsets, yielding *exclusive-between* code usage and clearer inter-group separation.

Together, these two regularizers impose diversity within each group and exclusivity between different groups. These constraints enhance the representational capacity of the VQ branch by ensuring better codebook utilization and preventing codebook collapse. The overall objective is incorporated as an additional regularizer alongside the VQ reconstruction, commitment, and codebook losses, further improving the quantization and learning process.

### 3.5. Training Strategies

**Total Learning Objective.** We supervise the continuous and discrete branches with standard objectives and combine them with lightweight regularizers. We also adopt a classical objective for VAE reconstruction branch that mixes pixel fidelity, perceptual similarity, KL prior and adversarial loss:

$$
\begin{aligned}
\mathcal{L}_{\mathrm{VAE}} = {} & \lambda_{\mathrm{pix}} \, \mathcal{L}_{\mathrm{vae\text{-}rec}}\big(\hat{X}^{(\mathrm{vae})}, X\big) \\
& + \lambda_{\mathrm{perc}} \, \mathcal{L}_{\mathrm{perc}}\big(\{\phi_\ell(\hat{X}^{(\mathrm{vae})})\}_\ell, \{\phi_\ell(X)\}_\ell\big) \\
& + \beta \, \mathcal{L}_{\mathrm{KL}}\big(q_\varphi(Z_K^{(\mathrm{vae})} \mid X), \, p(Z)\big) \\
& + \lambda_{\mathrm{gan}} \, \mathcal{L}_{\mathrm{gan}}^G\big(\hat{X}^{(\mathrm{vae})}\big).
\end{aligned} \quad (18)
$$

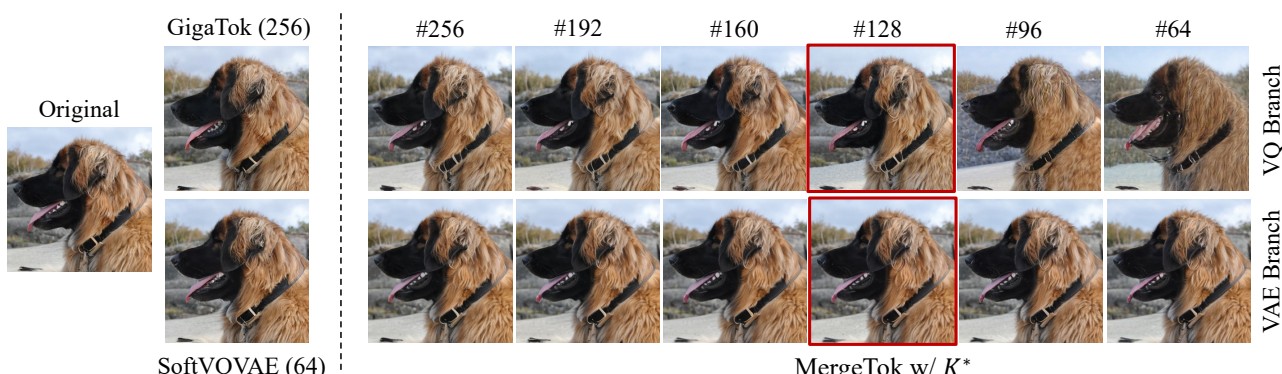

*Figure 4.* **Reconstruction in both VQ and VAE branches across Token Granularities.** We visualize reconstructions from both branches while sweeping the target sampling center $K^*$ controlled by merge ratio $r$ from #256 (left) down to #64 (right). It shows MergeTok's robustness to varying compression rates. The **red marker** indicates the optimal kept-token count ($K^* = 128$) identified during training.

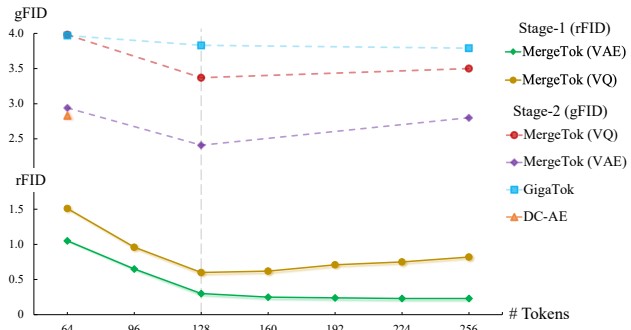

*Figure 5.* **Kept token number *vs* rFID/gFID** on ImageNet-256. With the optimal kept-token count of VAE branch $K^* = 128$, our MergeTok can achieve competitive rFID and gFID simultaneously.

where $\phi_\ell(\cdot)$ are fixed perceptual features and $\beta > 0$ controls KL strength.

For the VQ branch we use the standard reconstruction, commitment, and codebook update terms:

$$\mathcal{L}_{\text{VQ}} = \lambda_{\text{rec}}\,\mathcal{L}_{\text{vq-rec}}\big(\hat{X}^{(\text{vq})}, X\big) + \lambda_{\text{cb}}\,\mathcal{L}_{\text{codebook}}\big(\tilde{Z}_L^{(\text{vq})}, \mathcal{C}\big)$$
$$+ \lambda_{\text{com}}\,\mathcal{L}_{\text{com}}\big(Z_L^{(\text{vq})}, \tilde{Z}_L^{(\text{vq})}\big) + \lambda_{\text{gan}}\,\mathcal{L}_{\text{gan}}^G\big(\hat{X}^{(\text{vq})}\big). \tag{19}$$

The total objective is a weighted sum of the branch losses and three auxiliary regularizers:

$$\mathcal{L}_{\text{total}} = \lambda_{\text{vae}}\mathcal{L}_{\text{VAE}} + \lambda_{\text{vq}}\mathcal{L}_{\text{VQ}} + \lambda_{\text{ali}}\mathcal{L}_{\text{align}}$$
$$+ \lambda_{\text{div}}\mathcal{L}_{\text{div}} + \lambda_{\text{cons}}\mathcal{L}_{\text{cons}}. \tag{20}$$

where $\{\lambda_*\}$ are nonnegative loss weights. $\mathcal{L}_{\text{align}}$ imposes alignment on merged tokens, $\mathcal{L}_{\text{div}}$ encourages diverse group-wise code usage, $\mathcal{L}_{\text{cons}}$ promotes assignment consistency.

**Dynamic Sampling of Merge Ratios.** To improve the robustness of the VAE decoder to varying token granularities, we perform dynamic sampling to the token merging ratio during training. Specifically, at each training step, we discretely sample the number of retained tokens

$K \in \{k_1, k_2 \ldots, k_t\}$ from a truncated Gaussian distribution with mean $\mu$ and standard deviation $\sigma$, written as:

$$k \sim \text{Discrete-}\mathcal{N}(\mu = K^*, \sigma^2), \ \ \text{s.t. } k \in \{k_1, \ldots, k_t\}, \tag{21}$$

where $K^*$ denotes a hyperparameter that approximates the dataset's *average information density*, and $\{k_i\}$ enumerates the admissible kept-token counts; $\sigma$ controls the dispersion of the discrete Gaussian, and sampling is clipped to the valid set. The corresponding merge ratio is then computed by a scheduling function, $r = \text{merge\_ratio\_gernerator}(K, \beta)$, which determines the optimal merge ratio $r$ given the retained token count $K$ and a decay factor $\beta$. This exposes the model to varied token granularities, promoting generalization across compression levels and improving the decoder's adaptability to latents. We show the reconstruction result of the VQ and VAE branch with different sampled merge ratios in Fig. 4, and the rFID and gFID of different methods with different remaining tokens in Fig. 5.

## 4. Experiments

### 4.1. Implementation Details

**Visual Tokenizer Setup.** We offer two parameterizations of MergeTok, denoted SB and BL. The SB configuration employs an attention encoder with approximately 19M parameters, 6 blocks, 8 attention heads, 512-dimensional embeddings, and an attention decoder with approximately 86M parameters, 6 blocks, 12 heads, and 768-dimensional embeddings. The BL configuration adopts a base attention encoder with roughly 86M parameters, while the attention decoder is enlarged to about 329M parameters with 24 blocks, 16 heads, and 1024-dimensional embeddings. We employ a frozen DINOv2 ViT-B as the teacher in the alignment branch. The VQ branch uses a codebook with 16,384 entries and an 8-dimensional code embedding. During training, we adopt a token-merging schedule centered at 128 kept tokens; specifically, we sample the retained token

*Table 1.* **System-level comparison of discrete tokenizers on ImageNet 256×256.** We compare MergeTok against various auto-regressive models for the discrete tokenizers across three tasks: reconstruction, representation learning, and class-conditional generation. We report rFID for reconstruction, while gFID and IS for generation. Top-1 accuracy of linear probing (Lin.) is reported for learned representation. "#Tokens" denotes the latent token length, "#Code" means the codebook size, and "Ratio" is the fixed downsampling ratio. ⋆ denotes training tokenizers with frozen DINO discriminator for better reconstruction, and "CFG" denotes using classifier-free guidance.

| Tokenizer Method | Date | Ratio | Type | #Tokens | # Code | Lin. Acc. | rFID | Generator Method | #Param. | #Step | w/o CFG gFID↓ | IS↑ | w/ CFG gFID↓ | IS↑ |
|---|---|---|---|---|---|---|---|---|---|---|---|---|---|---|
| Taming-VQGAN (Esser et al., 2021) | CVPR'21 | 16 | 2D VQ | $16^2$ | $2^{10}$ | – | 7.94 | Taming-Trans. | 1.4B | 256 | 15.78 | 78.3 | – | – |
| RQ-VAE (Lee et al., 2022) | CVPR'22 | 16 | 2D RQ | $16^2$ | $2^{10}$ | – | 3.20 | RQ-Trans.-re | 1.4B | 64 | 8.71 | 119.0 | – | – |
| ViT-VQGAN (Yu et al., 2021) | ICLR'21 | 8 | 2D VQ | $32^2$ | $2^{13}$ | 65.1 | 1.28 | VIM-L | 1.7B | 1024 | 4.17 | 175.1 | – | – |
| MaskGIT (Chang et al., 2022) | CVPR'22 | 16 | 2D VQ | $16^2$ | $2^{10}$ | 57.4 | 2.28 | MaskGIT | 227M | 8 | 6.18 | 182.1 | – | – |
| LlamaGen (Sun et al., 2024) | NIPS'24 | 16 | 2D VQ | $16^2$ | $2^{14}$ | 47.6 | 2.19 | LlamaGen-L | 343M | 256 | 3.80 | 248.3 | 3.07 | 256.1 |
| LlamaGen (Sun et al., 2024) | NIPS'24 | 16 | 2D VQ | $16^2$ | $2^{14}$ | 47.6 | 2.19 | LlamaGen-XL | 775M | 256 | 3.39 | 227.1 | 2.62 | 244.1 |
| LlamaGen (Sun et al., 2024) | NIPS'24 | 16 | 2D VQ | $16^2$ | $2^{14}$ | 47.6 | 2.19 | LlamaGen-XXL | 1.4B | 256 | – | – | 2.34 | 253.9 |
| OmniTokenizer (Wang et al., 2024) | NIPS'24 | 16 | 2D VQ+VAE | $16^2$ | $2^{13}$ | – | 1.11 | GPT2 | 650M | 256 | 7.45 | 146.7 | – | – |
| VAR⋆ (Tian et al., 2024) | NIPS'24 | – | 2D RQ | 680 | $2^{12}$ | – | 0.90 | VAR-d16 | 310M | 10 | – | – | 3.30 | 274.4 |
| VAR⋆ (Tian et al., 2024) | NIPS'24 | – | 2D RQ | 680 | $2^{12}$ | – | 0.90 | VAR-d24 | 1.0B | 10 | – | – | **2.09** | **312.9** |
| VFMTok (Zheng et al., 2025b) | NIPS'25 | 16 | 2D VQ | $16^2$ | $2^{14}$ | 69.4 | 0.89 | LlmaGen-B | 111M | 256 | 3.09 | 173.6 | 3.43 | 252.2 |
| VFMTok (Zheng et al., 2025b) | NIPS'25 | 16 | 2D VQ | $16^2$ | $2^{14}$ | 69.4 | 0.89 | LlmaGen-L | 343M | 256 | 2.11 | 230.1 | 2.75 | 278.8 |
| VFMTok (Zheng et al., 2025b) | NIPS'25 | 16 | 2D VQ | $16^2$ | $2^{14}$ | 69.4 | 0.89 | LlamaGen-XXL | 1.4B | 256 | **1.95** | 259.3 | 2.19 | 278.0 |
| UniTok⋆ (Ma et al., 2025) | NIPS'24 | 16 | 2D MSQ | $16^2$ | $2^{15}$ | 70.8 | **0.41** | LlamaGen-XXL | 1.4B | 256 | 2.51 | 216.7 | 2.77 | 227.5 |
| MAGVITv2 (Yu et al., 2024a) | NIPS'24 | 16 | 2D LFQ | $16^2$ | $2^{18}$ | – | 0.98 | MAGVITv2 | 307M | 64 | 3.65 | 200.5 | 1.78 | 319.4 |
| B-AE-d20 (Hao et al., 2025) | ICLR'25 | 16 | 2D LFQ | 256 | $2^{20}$ | 67.5 | – | BiGR-L-d24 | 336M | 256 | – | – | 2.71 | 275.7 |
| B-AE-d32 (Hao et al., 2025) | ICLR'25 | 16 | 2D LFQ | 256 | $2^{32}$ | 69.8 | 1.69 | BiGR-XL-d32 | 799M | 256 | – | – | 2.49 | 278.8 |
| OpenMAGVIT2 (Luo et al., 2024) | arXiv'24 | 16 | 2D LFQ | $16^2$ | $2^{18}$ | – | 1.17 | LlamaGen-B | 343M | 256 | 3.08 | 258.3 | – | – |
| OpenMAGVIT2 (Luo et al., 2024) | arXiv'24 | 16 | 2D LFQ | $16^2$ | $2^{18}$ | – | 1.17 | LlamaGen-XL | 1.5B | 256 | 2.33 | 271.8 | – | – |
| MaskBIT (Weber et al., 2024) | TMLR'24 | 16 | 2D LFQ | $16^2$ | $2^{14}$ | – | 1.61 | MaskBIT | 305M | 64 | – | – | 1.65 | **341.8** |
| IBQ (Shi et al., 2024) | ICCV'25 | 16 | 2D LFQ | $16^2$ | $2^{18}$ | – | 1.00 | LlamaGen-B | 342M | 64 | 2.88 | 254.7 | – | – |
| IBQ (Shi et al., 2024) | ICCV'25 | 16 | 2D LFQ | $16^2$ | $2^{18}$ | – | 1.00 | LlamaGen-XL | 1.1B | 64 | **2.14** | **279.0** | – | – |
| FlowMo (Sargent et al., 2025) | ICCV'25 | 16 | 2D Diff+LFQ | $16^2$ | $2^{18}$ | – | **0.95** | LlamaGen-B | 397M | 256 | 4.30 | 274.0 | – | – |
| TokenBridge (Wang et al., 2025) | ICCV'25 | 16 | 2D VAE+LFQ | $16^2$ | $16^{64}$ | – | 1.11 | MAR-L | 486M | 256 | – | – | 1.76 | 294.8 |
| TokenBridge (Wang et al., 2025) | ICCV'25 | 16 | 2D VAE+LFQ | $16^2$ | $16^{64}$ | – | 1.11 | MAR-H | 910M | 256 | – | – | **1.55** | 313.3 |
| Titok-S-128 (Yu et al., 2024b) | NIPS'24 | 16 | 1D VQ | 128 | $2^{12}$ | 46.6 | 1.71 | MaskGIT-UViT-L | 287M | 8 | 4.61 | 166.7 | 2.50 | 278.7 |
| Titok-B-64 (Yu et al., 2024b) | NIPS'24 | 16 | 1D VQ | 64 | $2^{12}$ | 53.9 | 1.70 | MaskGIT-VIT | 177M | 8 | 3.08 | 192.5 | 2.48 | 214.7 |
| Titok-L-32 (Yu et al., 2024b) | NIPS'24 | 16 | 1D VQ | 32 | $2^{12}$ | 60.0 | 2.21 | MaskGIT-VIT | 177M | 8 | 3.15 | 173.0 | 2.77 | 199.8 |
| MergeVQ-GR (Li et al., 2025) | CVPR'25 | 16 | 1D LFQ | 144 | $2^{18}$ | 77.9 | 1.48 | RandAR-L | 343M | 64 | – | – | 2.63 | 279.5 |
| MergeVQ-GR (Li et al., 2025) | CVPR'25 | 16 | 1D LFQ | 256 | $2^{18}$ | 77.9 | 1.12 | MergeAR-L | 343M | 256 | 3.25 | 253.8 | – | – |
| GigaTok-SB (Xiong et al., 2025) | ICCV'25 | 16 | 1D VQ | 256 | $2^{14}$ | 61.5 | 0.89 | LlamaGen-B | 111M | 256 | – | – | 3.83 | 233.3 |
| GigaTok-BL⋆ (Xiong et al., 2025) | ICCV'25 | 16 | 1D VQ | 256 | $2^{14}$ | 64.1 | 0.51 | LlamaGen-B | 111M | 256 | – | – | 3.33 | 265.4 |
| GigaTok-BL (Xiong et al., 2025) | ICCV'25 | 16 | 1D VQ | 256 | $2^{14}$ | 63.8 | 0.81 | LlamaGen-XXL | 1.4B | 256 | 2.03 | 238.5 | – | – |
| Hita (Zheng et al., 2025a) | ICCV'25 | 16 | 1D VQ | 569 | $2^{14}$ | 36.6 | 1.03 | LlamaGen-B | 111M | 569 | – | – | 4.33 | 238.9 |
| Hita (Zheng et al., 2025a) | ICCV'25 | 16 | 1D VQ | 569 | $2^{14}$ | 36.6 | 1.03 | LlamaGen-L | 343M | 569 | – | – | **2.86** | 267.3 |
| **MergeTok-SB** | **Ours** | 16 | 1D VAE+VQ | 256 | $2^{14}$ | 73.8 | 0.97 | LlamaGen-B | 111M | 256 | 3.92 | 182.7 | 3.37 | 245.7 |
| **MergeTok-BL⋆** | **Ours** | 16 | 1D VAE+VQ | 256 | $2^{14}$ | 78.2 | **0.50** | LlamaGen-B | 111M | 256 | 3.56 | 247.6 | 3.09 | 267.8 |
| **MergeTok-BL** | **Ours** | 16 | 1D VAE+VQ | 256 | $2^{14}$ | **78.3** | 0.78 | LlamaGen-XXL | 1.4B | 256 | **1.93** | 265.4 | 2.14 | **281.5** |

numbers from $\{96, 128, 160, 196, 224, 256\}$. All models are trained on ImageNet-1K at 256×256 resolution for 200 epochs with a global batch size of 256. AdamW is adopted with a base learning rate of 1e-4, a cosine decay schedule, and momentum $(\beta_1, \beta_2) = (0.9, 0.95)$. Unless otherwise noted, the total objective is a weighted sum of the VQ loss, VAE loss, alignment loss, the intra-group diversity regularizer, and the inter-group consistency regularizer, with coefficients $1.0, 1.0, 0.5, 0.05$, and $0.05$, respectively.

**Visual Generator Setup.** *(i) VQ Branch:* We train two LlamaGen decoders: LlamaGen-B (111M parameters, 12 blocks, 12 heads) and LlamaGen-XXL (1.4B parameters, 48 blocks, 24 heads), and employ WSD learning rate scheduler with $1 \times 10^{-4}$ base rate, a decay ratio of 0.2, and a 1-epoch warm-up. AdaLN (Peebles & Xie, 2023) is not applied as it targets class-conditional setups. The batch size is 256

for LlamaGen-B/L and 512 for the XXL. All AR models are trained for 300 epochs on ImageNet-1K training set in $256 \times 256$. As for classifier guidance, when reporting gFID, we search the optimal guidance scale by a zero-order sweep with a step of 0.25 and report the lowest gFID obtained. *(ii) VAE Branch:* We adopt DiT and SiT as denoising models on ImageNet-1K at $256 \times 256$. Both models use a patch size of 1 and 1D absolute positional embeddings. We train the XL variants (675M) of DiT and SiT for 3M optimization steps and follow official configurations for other hyperparameters. For lighter ablations, we train SiT-L for 400K steps. All the protocols match prior works unless otherwise noted.

### 4.2. Comparison Results

**Discrete Methods on ImageNet-256.** Table 1 shows ImageNet-1K $256 \times 256$ results against discrete tokenizers

*Table 2.* **System-level comparison of continuous tokenizers on ImageNet 256×256**. We compare MergeTok against continuous methods on two tasks: reconstruction and class-conditional generation. We report rFID for reconstruction and use gFID and IS for generation. "#Tokens" is the latent token length, and "Down ratio" denotes the fixed downsampling ratio. "CFG" means using classifier-free guidance.

| Tokenizer Method | Date | Type | #Tokens | Down. ratio | Tok. rFID↓ | Generator Method | Training Epochs | #Param. | w/o CFG gFID↓ | w/o CFG IS↑ | w/ CFG gFID↓ | w/ CFG IS↑ |
|---|---|---|---|---|---|---|---|---|---|---|---|---|
| SD-VAE (Kingma, 2013) | ICLR'13 | 2D VAE | $32^2$ | 8 | 0.61 | DiT-XL/2 | 1400 | 675M | 9.62 | 121.5 | 2.27 | 278.2 |
| SD-VAE (Kingma, 2013) | ICLR'13 | 2D VAE | $32^2$ | 8 | 0.61 | SiT-XL/2 | 1400 | 675M | – | – | 2.62 | 252.2 |
| REPA (Yu et al., 2025) | ICLR'25 | 2D VAE | $32^2$ | 8 | 0.61 | DiT-XL/2 | 800 | 675M | 5.78 | 158.3 | **1.29** | 306.3 |
| DC-AE-f32 (Chen et al., 2025c) | ICLR'25 | 2D VAE | $8^2$ | 32 | 0.69 | DiT-XL/1 | 500 | 675M | 9.56 | – | 2.84 | – |
| DC-AE-f32 (Chen et al., 2025c) | ICLR'25 | 2D VAE | $8^2$ | 32 | 0.69 | SiT-XL/1 | 500 | 675M | 7.47 | – | 2.41 | – |
| VAVAE (Yao & Wang, 2025) | CVPR'25 | 2D VAE | $16^2$ | 16 | 0.61 | LightningDiT-XL/1 | 800 | 675M | 2.17 | 205.6 | 1.35 | 295.3 |
| RAE (DINOv2-S) (Zheng et al., 2025c) | arXiv'25 | 1D VAE | 256 | 16 | **0.49** | DiT-XL/1 | 800 | 675M | **1.87** | **209.7** | 1.41 | **309.4** |
| TiTok-BL-KL (Yu et al., 2024b) | NIPS'24 | 1D VAE | 64 | 16 | 1.25 | SiT-L/1 | 400 | 458M | 23.35 | 54.7 | – | – |
| MAR (Li et al., 2024) | NIPS'24 | 2D VQ | 256 | 16 | 1.22 | MAR-H/1 | 800 | 943M | 2.35 | 227.8 | 1.55 | 303.7 |
| SoftVQ-S⋆(Chen et al., 2025b) | CVPR'25 | 1D VQ | 256 | 16 | 0.80 | SiT-L/1 | 400 | 458M | 9.21 | 93.6 | – | – |
| SoftVQ-BL⋆(Chen et al., 2025b) | CVPR'25 | 1D VQ | 64 | 16 | 0.65 | DiT-XL/1 | 300 | 675M | 6.53 | 131.9 | 3.11 | 268.3 |
| SoftVQ-BL⋆(Chen et al., 2025b) | CVPR'25 | 1D VQ | 64 | 16 | 0.65 | SiT-XL/1 | 300 | 675M | 5.80 | 143.5 | 1.88 | 287.9 |
| SoftVQ-L⋆(Chen et al., 2025b) | CVPR'25 | 1D VQ | 64 | 16 | 0.61 | SiT-XL/1 | 300 | 675M | 5.35 | 151.2 | 1.86 | 293.6 |
| **MergeTok-BL** | **Ours** | 1D VAE+VQ | 256 | 16 | **0.48** | DiT-XL/1 | 800 | 675M | 1.91 | 211.4 | 1.44 | 304.1 |
| **MergeTok-BL** | **Ours** | 1D VAE+VQ | 256 | 16 | **0.48** | SiT-XL/1 | 800 | 675M | **1.79** | **217.6** | **1.29** | **311.7** |

*Table 3.* **Ablation of dual-branch architecture** with ToMe for the VQ and VAE reconstruction built upon the GigaTok baseline.

| Model | VQ | VAE | ToMe | Align | VAE-rFID | VQ-rFID |
|---|---|---|---|---|---|---|
| GigaTok-SB | ✓ | ✗ | ✗ | ✗ | – | 1.12 |
| | ✗ | ✓ | ✗ | ✗ | 0.81 | – |
| | ✓ | ✓ | ✗ | ✗ | 0.83 | 1.06 |
| | ✗ | ✓ | ✓ | ✗ | 0.75 | – |
| | ✓ | ✓ | ✓ | ✗ | 0.67 | 1.01 |
| | ✗ | ✓ | ✓ | ✓ | 0.63 | – |
| MergeTok-SB | ✓ | ✓ | ✓ | ✓ | 0.59 | 0.96 |

*Table 4.* **Ablation of semantic and group-aware loss constraints** based on MergeTok-SB for both the VQ and VAE branches.

| Group Constraints | Alignment | Entropy | VAE-rFID | VQ-rFID |
|---|---|---|---|---|
| ✓ | ✗ | ✗ | 0.78 | 1.01 |
| ✗ | ✓ | ✗ | 0.65 | 1.08 |
| ✗ | ✗ | ✓ | 0.77 | 1.09 |
| ✓ | ✓ | ✓ | 0.62 | 1.02 |
| ✓ | ✓ | ✗ | 0.59 | 0.97 |

in auto-regressive generation. Without a DINO discriminator, MergeTok attains 0.97 rFID with the SB configuration and 0.78 rFID with the BL one, outperforming comparable VQ baselines. With a DINO discriminator, the BL variant further improves to 0.50 rFID, establishing a new state of the art. For class-conditional image generation, with Llama-Gen as a second-stage generator, our XXL setup achieves 1.93 gFID without CFG, surpassing competitive tokenizers.

**Continuous Methods on ImageNet-256.** To verify the VAE branch, Table 2 offers comparisons of continuous tokenizers with reconstruction and generation results. Merge-Tok BL configuration hits a reconstruction FID of 0.48, outperforming contemporary continuous visual tokenizers such as DC-AE and SoftVQ under comparable settings. For downstream class-conditional generation, pairing our tokenizer with DiT yields gFID 1.91, while with SiT yields gFID 1.79, which indicates strong compatibility with diffusion generators and competitive end-to-end fidelity.

### 4.3. Ablation Study

We examine several key design choices on the MergeTok-SB model. Table 3 shows that adding semantic constraints via ToMe to the VAE branch reduces VAE-rFID from a baseline of 0.81 to 0.67. Including alignment loss improves VAE-rFID to 0.63. The full MergeTok-SB, which jointly optimizes both VAE and VQ branches, achieves the best VAE-rFID of 0.59, VQ-rFID of 0.96, validating that coupling continuous and discrete paths can lead to complementary gains. Table 4 ablates the loss components. Group constraints alone lift VQ rFID to 1.01. Adding alignment further improves the VQ branch and stabilizes the continuous path. Entropy loss exhibits a performance drop when combined with group constraints. Therefore, we choose the combination of group constraints and alignment.

## 5. Conclusion

We introduce *MergeTok*, a unified visual tokenizer bridging continuous (VAE) and discrete (VQ) modeling via token merging. The merging provides a semantic conduit, enabling (i) merged-token alignment to regularize the VAE latent space; and (ii) group-aware quantization to stabilize VQ training and improve codebook utilization. A shared encoder-decoder is trained end-to-end with a single joint objective and light merge-ratio sampling. On ImageNet-1K at $256 \times 256$, *MergeTok* substantially improves rFID over continuous-only and discrete-only baselines, producing compact sequences for both AR and diffusion generators. This shows that a single, unified architecture can be simultaneously semantics-aware and generator-friendly.

## Impact Statement

This paper aims to advance the field of visual generation by proposing a unified visual tokenizer, MergeTok, which integrates continuous and discrete representations through token merging. Our method provides a more stable and efficient way to learn semantic token representations, potentially benefiting downstream tasks such as image generation, compression, and multimodal understanding. While Merge-Tok is primarily intended for research purposes, we do not foresee any immediate societal consequences that need to be specifically highlighted.

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

# A. Token Merging and Unmerging

## A.1. Token Merging in Encoding

Following ToMe (Bolya et al., 2023), MergeTok inserts a lightweight merge unit into each Transformer encoder block to progressively reduce the number of spatial tokens while preserving semantic content. Given the patch-token sequence $Z_L \in \mathbb{R}^{L \times D}$, ToMe performs merging in four steps:

1. Evenly partition tokens into two sets $A$ and $B$ (e.g., by odd/even indices).

2. For each token in $A$, find its most similar token in $B$ according to the current-layer attention features

3. Select the top pairs for merging under a pre-defined schedule

4. Aggregate features within each selected pair (e.g., by averaging) to form merged tokens.

After one merge operation, the sequence length is reduced from $L$ to $K$ ($K < L$), producing a compressed sequence $Z_K \in \mathbb{R}^{K \times D}$ that is fed into the next layer. Token similarity is computed using attention features from the current layer, typically the self-attention keys Key to match tokens across $A$ and $B$. Since each merged token may represent multiple original tokens, ToMe tracks the *token size* $s$ (the number of originals aggregated into a surviving token) and compensates its influence in self-attention by adding $\log s$ to the attention logits:

$$\mathbf{A} = \mathrm{softmax}\left(\frac{Q \, \mathrm{Key}^\top}{\sqrt{d}} + \log s\right), \qquad (22)$$

where $Q$ and Key denote the query and key matrices, $d$ is the attention head dimension, and $s$ is broadcast to match the logits shape. Across layers, ToMe follows a merging schedule that controls how many pairs are merged, trading off efficiency and fidelity. In practice, we adopt a decreasing schedule (e.g., square decreasing) where the number of merges increases with depth to encourage stronger semantic abstraction at higher layers.

## A.2. Source Map For Token Unmerging

In this subsection, we provide a concrete example to illustrate how token merging and the subsequent unmerging process can be expressed using the *source map* $S$ and its one-hot matrix form $A$.

As introduced in Sec. 3.3, the VAE branch performs token merging during encoding: it compresses the original length-$L$ token sequence $Z_L \in \mathbb{R}^{L \times D}$ into a compact sequence $Z_K^{(\mathrm{vae})} \in \mathbb{R}^{K \times D}$, accompanied by a discrete source map $S$ (cf. Eq. 2). During decoding, the reconstruction process conditions on both $Z_K^{(\mathrm{vae})}$ and $S$ to recover the original structure (cf. Eq. 3). Each entry $S_i$ (for $i = 1, \ldots, L$) indicates the index of the merged token to which the $i$-th original token belongs.

**Example.** Consider an example with $L = 5$ and $K = 3$. Let the original token sequence be denoted as $Z_L = [z_1, z_2, z_3, z_4, z_5]^\top \in \mathbb{R}^{5 \times D}$, where each $z_i \in \mathbb{R}^D$ is a token embedding. Suppose the source map is given by:

$$S = [0, 0, 1, 1, 2], \qquad (23)$$

indicating that the first two tokens are assigned to cluster 0, the next two to cluster 1, and the last token to cluster 2.

This assignment can be encoded by a one-hot *source matrix* $A \in \{0,1\}^{L \times K}$, where each row $i$ has a 1 in column $S_i$:

$$A_{ij} = \begin{cases} 1, & \text{if } S_i = j \\ 0, & \text{otherwise} \end{cases} \qquad (24)$$

For the given $S$, the source matrix becomes:

$$A = \begin{pmatrix} 1 & 0 & 0 \\ 1 & 0 & 0 \\ 0 & 1 & 0 \\ 0 & 1 & 0 \\ 0 & 0 & 1 \end{pmatrix} \in \{0,1\}^{5 \times 3} \qquad (25)$$

**Token merging.** Given the original sequence $Z_L$, the compressed sequence of merged tokens $Z_K = [z_0^{(\mathrm{m})}, z_1^{(\mathrm{m})}, z_2^{(\mathrm{m})}]^\top \in \mathbb{R}^{K \times D}$ can be computed by:

$$Z_K = A^\top Z_L \qquad (26)$$

In this case:

$$z_0^{(\mathrm{m})} = \frac{z_1 + z_2}{2}, \quad z_1^{(\mathrm{m})} = \frac{z_3 + z_4}{2}, \quad z_2^{(\mathrm{m})} = z_5 \quad (27)$$

**Unmerging.** The reconstruction restores the token sequence layout by broadcasting each merged token back to its original positions:

$$\hat{Z}_L = A Z_K. \qquad (28)$$

For this example:

$$\hat{Z}_L = A Z_K = \begin{pmatrix} 1 & 0 & 0 \\ 1 & 0 & 0 \\ 0 & 1 & 0 \\ 0 & 1 & 0 \\ 0 & 0 & 1 \end{pmatrix} \begin{pmatrix} (z_0^{(\mathrm{m})})^\top \\ (z_1^{(\mathrm{m})})^\top \\ (z_2^{(\mathrm{m})})^\top \end{pmatrix} = \begin{pmatrix} (z_0^{(\mathrm{m})})^\top \\ (z_0^{(\mathrm{m})})^\top \\ (z_1^{(\mathrm{m})})^\top \\ (z_1^{(\mathrm{m})})^\top \\ (z_2^{(\mathrm{m})})^\top \end{pmatrix} \qquad (29)$$

*Figure A1.* **Illustration of existing tokenizers and our MergeTok tokenizer. (a) Naive VQ tokenization**, which performs codebook look-up and dictionary learning to get quantized embedding sequences to enable auto-regressive generation. **(b) Naive VAE tokenization**, which maps and samples the latent tokens as a mixture of Gaussian distributions to facilitate latent diffusion generation. **(c) Our MergeTok tokenizer** has two branches with the shared Encoder and Decoder. During training, we first forward the VAE branch (with ToMe) to get merged continuous sequences in $K \times d$, which will be reparameterized through diagonal Gaussian distributions and aligned to condensed semantic tokens from the teacher model DINOv2 (with ToMe). Then, we forward the VQ branch (without ToMe) to get quantized embedding sequences in $L \times d$, where the codebook commit loss and our proposed Group Constraint losses will enable timely codebook update and great codebook usage. The ToMe operation and semantic constraints from the VAE branch can also enhance the clustering effects of the VQ codebook.

This demonstrates how the source map $S$ and its matrix form $A$ can be used to implement both token merging and un-merging using simple matrix multiplication. Importantly, $S$ and $A$ not only support efficient aggregation and broadcasting of features, but also retain the spatial correspondence between the original and merged tokens. Each entry in $S$ records the precise assignment of an original token to its semantic group, effectively encoding its relative spatial location within the compressed representation. This spatial mapping enables the design of a hybrid decoder architecture: after semantic tokens $Z_K^{(\text{vae})}$ are decoded (e.g., via a transformer decoder), the source map can be used to guide the unmerging process, distributing semantic content back to spatial positions in the form of $\hat{Z}_L$, which can then be further refined by a pixel-level decoder to reconstruct the image. In this way, the source map serves as a lightweight yet effective bridge between discrete semantic representations and fine-grained spatial reconstructions, facilitating high-fidelity decoding in both semantic and pixel domains.

## B. Implement Details

**MergeTok Models.** We provide two parameter scales for the MergeTok tokenizer. Based on GigaTok (Xiong et al., 2025), the SB configuration employs a small-size encoder coupled with a base-size decoder, while the BL configuration uses a base-size encoder and a large-size decoder. The detailed architectural specifications and training hyperparameters of both variants are reported in Table A1. For the autoregressive generator, we adopt Llama-Gen as the backbone architecture (Sun et al., 2024) and instantiate three model sizes, namely Base, Large, and XX-Large (XXL).

Their parameter counts and experimental settings are summarized in Table A2. For the diffusion generator, we adopt SiT and DiT as the backbone (Peebles & Xie, 2023; Ma et al., 2024). Our DiT and SiT generators closely follow the original architectural designs. For DiT, we adopt the standard Transformer-based diffusion backbone operating on 1D latent token sequences (Rombach et al., 2022), using 1D absolute positional encodings (Dosovitskiy et al., 2021) and keeping the depth, embedding dimension, number of attention heads, and MLP width identical to the corresponding DiT variants in the original work. Similarly, our SiT models reuse the same overall Transformer block structure as DiT while incorporating the shift-based operations introduced in the original SiT architecture, and are likewise applied to 1D latent token sequences with absolute positional encodings. Apart from adapting the input interface to our 1D latent representation, we do not introduce any additional architectural modifications to either DiT or SiT.

**Evaluation of Representation.** To measure the representation with semantics and contextual information, we follow previous self-supervised learning (He et al., 2022) and generative models (Li et al., 2023; 2025) to conduct the linear probing protocol, as shown in Table 1. The linear classification is performed upon the latent embedding space of trained encoders by fine-tuning a parameter-free BN layer and a linear layer for 90 epochs using the AdamW optimizer with a batch size of 1024. The basic learning rate is set to $1 \times 10^{-3}$ and advanced augmentations and training strategies for modern architectures (Touvron et al., 2021) will not be used.

*Table A1.* Implementation details and configuration of network architecture, hyperparameters of loss functions, and training settings for the two versions of MergeTok tokenizers on ImageNet-1K.

| Settings | MegreTok-SB | MergeTok-BL |
|---|---|---|
| Channels | 256 | 256 |
| CNN Stage number | 5 | 5 |
| Channel multiplier | [1, 1, 2, 2, 4] | [1, 1, 2, 2, 4] |
| Encoder Attention Blocks | 6 | 12 |
| Encoder Attention Heads | 8 | 12 |
| Encoder Attention Dim | 512 | 768 |
| Decoder Attention Blocks | 12 | 24 |
| Decoder Attention Heads | 12 | 16 |
| Decoder Attention Dim | 768 | 1024 |
| Vocabulary size | 16384 | 16384 |
| Discriminator loss | 0.5 | 0.5 |
| Perceptual loss | 1.0 | 1.0 |
| VQ rec loss | 1.0 | 1.0 |
| VAE rec loss | 1.0 | 1.0 |
| Diversity loss | 0.05 | 0.05 |
| Consistency loss | 0.05 | 0.05 |
| Commitment loss | 0.25 | 0.25 |
| Alignment loss | 0.5 | 0.5 |
| Optimizer | AdamW | AdamW |
| $(\beta_1, \beta_2)$ | (0.9, 0.95) | (0.9, 0.95) |
| Weight decay | 1e-4 | 0.0 |
| Training epochs | 200 | 200 |
| Base learning rate | 1e-4 | 1e-4 |
| Batch size | 256 | 256 |
| LR scheduler | cosine_v2 | cosine_v2 |
| #Param. of Attn Encoder | 19M | 86M |
| #Param. of Attn Decoder | 86M | 329M |

**Evaluation of Generation.** We evaluate the generative performance of MergeTok-based models on ImageNet-1K in 256×256 resolutions. As for reconstruction tasks, we follow VQGAN (Esser et al., 2021) to report the reconstruction Fréchet Inception Distance (rFID) computed on the 50K validation set with `CenterCrop`. It measures how well the continuous–discrete hybrid latent representation preserves the image distribution. As for class-condition image generation, we strictly follow the setup and use the same reference batches of ADM (Dhariwal & Nichol, 2021) with their official implementation and evaluate on a single H20-96G GPU and tf32 precision. We report generation FID (gFID), Inception Score (IS), as well as Precision and Recall to jointly characterize fidelity and diversity of the samples. The gFID measures the feature distance between the distributions of real and generated images on the Inception-v3 embeddings under the multivariate Gaussian distributions. IS also uses the Inception-v3 network, but uses the softmax-normalized logit for evaluation of the metric. Unless otherwise specified, all metrics are computed on 50K generated images and are compared against the ImageNet validation split. Following

common practice (Tian et al., 2024), we report results both with and without classifier-free guidance (CFG). For settings using CFG, we sweep the guidance scale on a held-out subset of the validation set and select the scale that yields the lowest gFID for each generator configuration.

## C. Visualization of Analysis

**Visualization of Reconstruction and Generation.** We further provide more visualization of reconstruction and generation of MergeTok tokenizers and relevant generators. We visualize the reconstruction results of both branches in the MergeTok-BL tokenizer on ImageNet-1k with 256×256 resolutions, as shown in Fig. A2. We also provide results of the continuous diffusion generators and the discrete auto-regressive generator built upon the VAE and VQ branches of our MergeTok tokenizer, respectively, as shown in Fig. A3.

**Experimental Analysis.** In Fig.A1, we provide a schematic overview of the MergeTok tokenization pipeline. The proposed design simultaneously addresses the respective limitations of both VQ and VAE tokenizers: for the VQ branch, we introduce continuous gradient updates to alleviate the intrinsic gradient discontinuity problem in VQ optimization, and employ a group-constraints loss to stabilize and regularize codebook learning; for the VAE branch, we impose semantic constraints that align features across branches, which not only mitigate the semantic disentanglement issues of standard VAEs but also facilitate more coherent clustering dynamics in the VQ codebook. The qualitative visualizations in Fig. **??** further substantiate these advantages. After applying semantic constraints, the VAE reconstructions exhibit highly recognizable structures with clear and salient features. In the VQ branch, benefiting from the guidance of the VAE branch, the quantized feature maps display stronger semantic interpretability than the original features; although the codebook lookup introduces slightly more mixed and dense semantic patterns compared to the VAE branch, the resulting representations still retain enhanced recognizability.

## D. Extended Related Work

**Advances in Visual Tokenizers for Image Generation.** Modern visual tokenizers convert images into discrete or continuous sequences to enable transformer-based generation. On the discrete side, VQ-based methods address gradient sparsity and codebook collapse by improving differentiability and scaling: IBQ propagates gradients over full code distributions to maintain high code utilization; LFQ replaces vector lookups with binary indices to unlock very large vocabularies; FQ factorizes a large codebook into coordinated sub-codebooks with disentanglement constraints and semantic guidance; and SoftVQ-VAE adopts

*Table A2.* Configuration of discrete and continuous visual generators with MergeTok for image generation on ImageNet-1K.

| Settings | LlamaGen-B | LlamaGen-L | LLamaGen-XXL | DiT-XL | SiT-XL |
|---|---|---|---|---|---|
| Attention heads | 12 | 24 | 48 | 16 | 16 |
| Input dim. | 768 | 1024 | 1536 | $16 \times 16 \times 4$ | |
| Num. layers | 12 | 24 | 1.4B | 28 | 28 |
| Dropout | 0.1 | 0.1 | 0.1 | 0 | 0 |
| Mask schedule | Arccos | Arccos | Arccos | – | – |
| Label smoothing | 0.1 | 0.1 | 0.1 | – | – |
| Sampler | – | – | – | Euler-Maruyama | |
| Gen. steps | 256 | 256 | 256 | 250 | 250 |
| # Parameter | 111M | 343M | 1.4B | 675M | 675M |
| Optimizer | AdamW | | | AdamW | |
| $(\beta_1, \beta_2)$ | (0.9, 0.95) | | | (0.9, 0.999) | |
| Weight decay | 5e-2 | 5e-2 | 5e-2 | 0 | 0 |
| Training epochs | 300 | 300 | 300 | 400 | 400 |
| Base learning rate | $1 \times 10^{-4}$ | $1 \times 10^{-4}$ | $1 \times 10^{-4}$ | $1 \times 10^{-4}$ | $1 \times 10^{-4}$ |
| Batch size | 256 | 256 | 512 | 256 | 256 |
| LR scheduler | WSD | WSD | WSD | – | – |

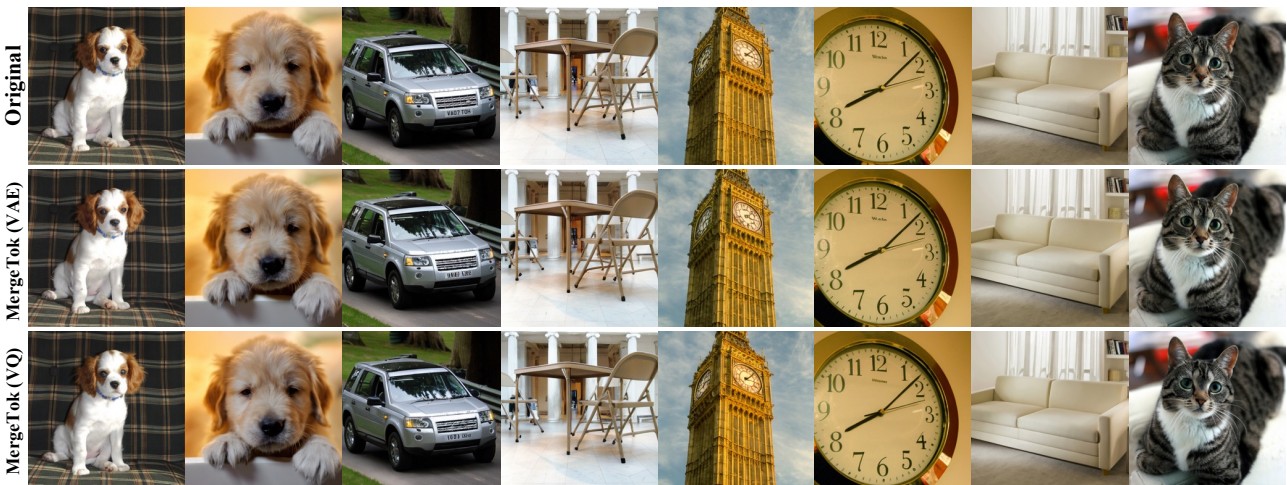

*Figure A2.* **Visualization of reconstruction** with the MergeTok-BL tokenizer on ImageNet-256. During inference time, we remove the ToMe modules employed during training from both the VAE and VQ branches to achieve the optimal reconstruction results.

soft categorical posteriors, yielding fully differentiable training and strong compression without sacrificing fidelity. On the continuous side, VAE-style tokenizers progress along two axes: high compression and semantic organization. DC-AE combines residualized latents with staged training to sustain quality at extreme spatial downsampling and to accelerate latent diffusion; MAETok shows that masked autoencoding can induce a more discriminative latent with far fewer tokens and substantially faster training; and representation alignment techniques (e.g., REPA and its end-to-end variant) align latent features to powerful visual encoders, stabilizing joint optimization with diffusion and improving both convergence and generative quality. Collectively, these advances expand vocabulary/capacity, restore gradient flow, and impose semantic structure, enabling tokenizers that are both reconstruction-faithful and modeling-friendly.

**Unified Visual Tokenizers for Multimodal and Multi-Task Systems.** Beyond standalone generation, recent work targets tokenizer designs that serve both recognition and synthesis, and even span modalities. UniTok argues that the perceived "loss conflict" between reconstruction and semantics stems from limited discrete capacity; multi-codebook quantization and wider embeddings mitigate this bottleneck, enabling low rFID and strong zero-shot accuracy from the same tokens while plugging seamlessly into MLLMs for native generation. AToken extends unification across images, video, and 3D with a transformer and 4D positional encoding, using adversarial-free perceptual/style objectives and a curriculum to achieve high-fidelity reconstruction and competitive downstream performance, thereby supporting cross-modal generation within a single token space. SPAE aligns vision with language by translating images into multi-scale lexical tokens consumable by frozen

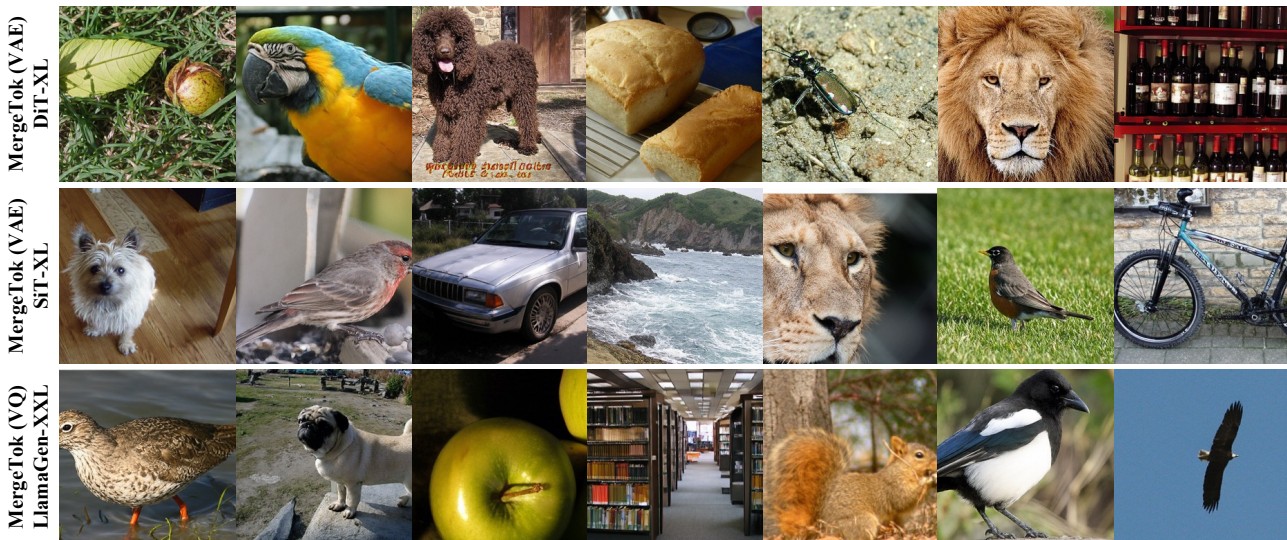

*Figure A3.* **Visualization of class-conditional generation** with different generators upon the MergeTok-BL tokenizer in ImageNet-256. Following REPA (Yu et al., 2025) and GigaTok (Xiong et al., 2025), we train DiT-XL and SiT-XL generators upon the VAE branch with full tokens (#256) while training a LlamaGen-XXL generator upon the VQ branch, where these generators achieve competitive generation performances.

LLMs, enabling both understanding and token-level image synthesis; similarly, unified tokenizers in autoregressive frameworks (e.g., MAGVIT-v2) show that with better visual tokens, GPT-style generators can rival or surpass diffusion on standard benchmarks. These systems illustrate that capacity scaling, multi-scale design, and language alignment are key to a single tokenizer that supports diverse tasks and modalities.

**Token Compression Techniques and Their Impact on Generative Models.** A complementary line of work reduces sequence length to improve efficiency and robustness, especially for autoregressive decoders where errors accumulate with depth. Adaptive-length tokenization recurrently distills 2D patches into compact 1D sequences, allocating more tokens to complex content and fewer to simple scenes, thereby matching capacity to entropy and reducing unnecessary decoding steps. Fixed aggressive compression via improved tokenizers (e.g., SoftVQ-VAE, DC-AE) achieves tens of tokens per image, yielding order-of-magnitude speedups with competitive fidelity. At inference, token-merging methods (e.g., ToMe and subsequent variants) dynamically fuse redundant tokens inside generative backbones to cut compute and memory with negligible perceptual loss; training-time designs (e.g., MergeVQ) integrate merging into the encoder to form short, quantized sequences while retaining mechanisms to recover fine detail at decode time. Together, adaptive token counts and principled merging reduce step count and memory pressure, often improving quality by limiting error propagation, and make large-scale, multimodal generation more tractable.

