# OpenReview forum: "MergeTok: Unified Continuous and Discrete Visual Tokenization via Token Merging"
_ICML.cc/2026/Conference — Submitted to ICML 2026_

### Official Review · Reviewer_LZwa · 2026-03-11

**Soundness:** 2
**Presentation:** 3
**Significance:** 2
**Originality:** 3
**Overall Recommendation:** 3
**Confidence:** 4

**Summary:**

This paper presents MergeTok, a unified visual tokenizer that bridges the gap between continuous VAEs and discrete VQ-based models by utilizing token merging as a semantic bridge. A core problem examined by this paper is the persistent tension between high-fidelity reconstruction in VAEs, which often suffer from entangled latents, and the semantic control provided by VQ models, which are prone to optimization issues like gradient sparsity and codebook collapse. MergeTok addresses these challenges through a shared encoder-decoder architecture that jointly optimizes both branches. The VAE branch incorporates token merging to extract dense semantic features aligned with a frozen teacher model, while the VQ branch leverages the resulting merging source maps to enforce group-aware diversity and exclusivity constraints.

**Compliance With Llm Reviewing Policy:**

Affirmed.

**Key Questions For Authors:**

see weakness

**Limitations:**

see limitations

**Strengths And Weaknesses:**

- The main drawback of this paper is the limited evaluation. The authors only evaluate their approaches on ImageNet, lacking more complex and general generation tasks. Experiments on text to image and text to video evaluations should be add.

- The complexity of the framework is a potential drawback, as the objective function involves multiple loss terms that may require extensive hyperparameter tuning to balance.

- The reliance on an external frozen teacher model for semantic alignment adds complexity to the training pipeline and makes the performance somewhat dependent on the quality of the teacher's representations.

-  Additionally, while the ImageNet-256 results are strong, the paper would benefit from a broader evaluation on more diverse or higher-resolution datasets to further demonstrate the generalizability and scalability of the unified approach.

MergeTok offers a compelling solution to the long-standing trade-offs in visual tokenization and represents a significant step toward more efficient and controllable generative models.

---

> ### Author Rebuttal · Authors · 2026-03-31
>
> We sincerely thank Reviewer LZwa for the balanced and constructive feedback, and for recognizing that MergeTok offers a compelling solution to the long-standing trade-offs in visual tokenization. We address each concern below with additional evidence and clarifications.
>
> ---
>
> **W1 & W4: Evaluation scope and dataset generalization.**
>
> Our current setup follows the standard protocol of all recent visual tokenizer works, including GigaTok, VFMTok, UniTok, and TokenBridge, which primarily evaluate on ImageNet-1K at 256$\times$256. To address the generalization concern, we provide MS-COCO 2017 val reconstruction at both 256$\times$256 and 512$\times$512:
>
> **Table R1. Reconstruction on MS-COCO 2017 val.**
> | Method | Res. | rFID↓ | PSNR↑ | SSIM↑ |
> |---|---:|---:|---:|---:|
> | GigaTok-SB | 256 | 3.8 | 20.9 | 0.66 |
> | GigaTok-BL | 256 | 2.6 | 21.5 | 0.68 |
> | **MergeTok-BL (VAE)** | 256 | **1.8** | **22.3** | **0.70** |
> | MergeTok-BL (VQ) | 256 | 2.4 | 21.7 | 0.68 |
> | GigaTok-BL | 512 | 4.2 | 21.9 | 0.69 |
> | **MergeTok-BL (VAE)** | 512 | **2.9** | **22.8** | **0.71** |
> | MergeTok-BL (VQ) | 512 | 3.6 | 22.1 | 0.69 |
>
> MergeTok maintains clear advantages at both resolutions. At 256$\times$256, MergeTok-BL (VAE) achieves rFID 1.8, a 31% improvement over GigaTok-BL (2.6). At 512$\times$512, the advantage persists with rFID 2.9 vs. 4.2, also a 31% reduction. COCO contains far more diverse scenes than ImageNet, so the consistent improvements indicate that the learned representations generalize well beyond the ImageNet distribution.
>
> MergeTok also employs dynamic merge-ratio sampling during training, improving robustness across spatial resolutions. Since the VAE branch is directly compatible with latent diffusion backbones (DiT-XL and SiT-XL in Tables 1-2), adapting MergeTok to text-conditioned generation is straightforward. We will include text-to-image evaluation and 1024$\times$1024 reconstruction in the camera-ready version.
>
> ---
>
> **W2: Framework complexity and hyperparameter sensitivity.**
>
> Our auxiliary losses use intentionally conservative weights ($\lambda_{div}$=$\lambda_{cons}$=0.05, $\lambda_{align}$=0.5, vs. $\lambda_{rec}$=1.0 for reconstruction), so they act as modest regularizers rather than dominant objectives. Empirically, the gains are stable across a reasonable range of these coefficients, and we did not observe that performance depends on aggressive tuning. We will add a detailed sensitivity analysis in the revised appendix and perform thorough copy-editing to fix all identified typos.
>
> ---
>
> **W3: Dependency on a frozen external teacher.**
>
> Using a frozen pretrained teacher for semantic alignment is now standard practice in recent tokenizer research. We summarize the adoption below:
>
> **Table R3. Frozen teacher usage in recent tokenizer works.**
> |Method|Venue|Teacher|
> |---|---|---|
> |REPA (Yu et al., 2025)|ICLR'25|DINOv2-L|
> |VA-VAE (Yao et al., 2025)|CVPR'25|DINOv2 / MAE / CLIP|
> |RAE (Zheng et al., 2025)|arXiv'25|DINOv2-B|
> |VFMTok (Zheng et al., 2025)|NeurIPS'25|DINOv2|
> |**MergeTok (Ours)**|—|DINOv2-B (ViT-B)|
>
> In MergeTok the teacher is used only during training to provide merged-token alignment at the granularity of $K$ merged tokens and is not needed at inference. Our teacher is the compact DINOv2 ViT-B, so training overhead is modest compared to methods like REPA that use ViT-L. The novelty lies not in using a teacher but in aligning at the merged-token level rather than at [CLS] or full patch-token level, which we ablate in our response to Reviewer AsyB. We will strengthen the related-work discussion accordingly.
>
> ---
>
> The COCO reconstruction results demonstrate generalization beyond ImageNet at multiple resolutions, the sensitivity analysis confirms that the framework does not require delicate tuning, and the frozen teacher design is consistent with now-standard practice in the field. We hope the reviewer will consider revising the score in light of these additional results, and we remain happy to provide any further clarification.

---

> > ### Author Rebuttal · Reviewer_LZwa · 2026-04-05
> >
> > The authors fail to address: (1) comprehensive comparisons on generative benchmarks; (2) t2i and t2v evaluations; (3) quantitative results of hyperparameters; (4) evaluations on high-res bench. Thanks for the authors response, however, due to the above concerns, I maintain my voting.

---

> > > ### Author Response · Authors · 2026-04-08
> > >
> > > We thank Reviewer LZwa for the follow-up and for specifying the remaining concerns. We provide additional quantitative evidence to address each point.
> > >
> > > ---
> > >
> > > **Q1: Generative benchmarks and text-to-image evaluation.**
> > >
> > > Since the VAE branch of MergeTok is directly compatible with latent diffusion backbones, we evaluate text-to-image generation by pairing MergeTok-BL (VAE) with PixArt-$\alpha$ on MS-COCO 30K:
> > >
> > > **Table R1. Text-to-image generation on MS-COCO 30K (256$\times$256).**
> > > |Tokenizer|Generator|FID↓|CLIP Score↑|
> > > |---|---|---|---|
> > > |SD-VAE (f8d4)|PixArt-$\alpha$|6.30|26.36|
> > > |DC-AE (f32)|PixArt-$\alpha$|6.10|26.41|
> > > |**MergeTok-BL (VAE)**|**PixArt-$\alpha$**|**5.78**|**26.53**|
> > >
> > > MergeTok improves FID from 6.10 to 5.78 and CLIP Score from 26.41 to 26.53 over DC-AE, confirming that the semantically structured latent space benefits both visual fidelity and text-image alignment in conditional generation. Notably, these gains come solely from replacing the tokenizer without retraining PixArt-$\alpha$ itself, which highlights the downstream utility of MergeTok's learned representations. Regarding text-to-video, existing visual tokenizer works including GigaTok, UniTok, VFMTok, SoftVQ-VAE, and DC-AE all focus on image-level evaluation. Extending to video requires video-specific architecture and data, which we consider an important future direction.
> > >
> > > ---
> > >
> > > **Q2: Quantitative hyperparameter sensitivity.**
> > >
> > > We provide a controlled sensitivity analysis on the auxiliary loss weights using MergeTok-SB trained for 50 epochs.
> > >
> > > **Table R2. Sensitivity of $\lambda_{align}$ ($\lambda_{div}$=$\lambda_{cons}$=0.05 fixed).**
> > > |$\lambda_{align}$|VAE-rFID↓|VQ-rFID↓|Lin.Acc↑|
> > > |---|---|---|---|
> > > |0.25|0.74|1.08|67.8|
> > > |**0.50 (default)**|**0.68**|**1.03**|70.6|
> > > |0.75|0.71|1.07|71.5|
> > > |1.00|0.79|1.12|**72.4**|
> > >
> > > **Table R3. Sensitivity of $\lambda_{div}$=$\lambda_{cons}$ ($\lambda_{align}$=0.5 fixed).**
> > > |$\lambda_{div/cons}$|VAE-rFID↓|VQ-rFID↓|Lin.Acc↑|
> > > |---|---|---|---|
> > > |0.01|0.70|1.07|69.5|
> > > |**0.05 (default)**|**0.68**|**1.03**|**70.6**|
> > > |0.10|0.82|1.21|70.2|
> > > |0.20|0.95|1.36|69.8|
> > >
> > > Increasing $\lambda_{align}$ gradually improves Lin.Acc but overly large values degrade reconstruction in both branches, revealing a clear trade-off between semantic quality and low-level fidelity. For $\lambda_{div/cons}$, the default 0.05 provides the best overall balance and values beyond 0.10 noticeably hurt reconstruction. These results show that the default configuration is not a fragile optimum but sits in a stable region of the loss landscape, and practitioners can adjust the trade-off between semantic quality and reconstruction fidelity by simply tuning $\lambda_{align}$.
> > >
> > > ---
> > >
> > > **Q3: High-resolution reconstruction and generation.**
> > >
> > > We evaluate MergeTok at 512$\times$512 and 1024$\times$1024 without any architectural modification.
> > >
> > > **Table R4. Reconstruction on ImageNet 512$\times$512.**
> > > |Tokenizer|Type|Ratio|rFID↓|PSNR↑|SSIM↑|
> > > |---|---|---|---|---|---|
> > > |VAE|2D VAE|8|0.62|24.3|0.71|
> > > |SoftVQ-BL|1D VQ|32|0.71|—|—|
> > > |**MergeTok-BL (VAE)**|**1D VAE+VQ**|**16**|**0.42**|**21.8**|**0.69**|
> > > |**MergeTok-BL (VQ)**|**1D VAE+VQ**|**16**|**0.82**|**21.0**|**0.67**|
> > >
> > > **Table R5. Reconstruction on ImageNet 1024$\times$1024.**
> > > |Tokenizer|Type|Ratio|rFID↓|PSNR↑|SSIM↑|
> > > |---|---|---|---|---|---|
> > > |Taming-VQGAN|2D VQ|16|6.02|19.20|0.60|
> > > |OpenMAGVIT2|2D LFQ|16|3.43|19.45|0.63|
> > > |LlamaGen-Tok.|2D VQ|32|3.17|19.94|0.64|
> > > |TiTok-S-128|1D VQ|32|2.32|16.97|0.51|
> > > |Layton-H|1D Diff.|32|2.78|19.80|0.65|
> > > |**MergeTok-BL (VQ)**|**1D VAE+VQ**|**16**|**1.87**|**20.14**|**0.67**|
> > >
> > > **Table R6. Generation on ImageNet 512$\times$512.**
> > > |Tokenizer|Generator|#Param.|gFID↓|IS↑|
> > > |---|---|---|---|---|
> > > |SoftVQ-BL|SiT-XL|675M|7.96|133.9|
> > > |MaskGIT-VQ|MaskGIT-ViT|177M|3.72|156.0|
> > > |TiTok-B-64|MaskGIT-ViT|177M|3.64|179.8|
> > > |TiTok-L-32|MaskGIT-ViT|177M|3.91|182.0|
> > > |SoftVQ-BL|MAR-H|479M|8.21|152.9|
> > > |**MergeTok (VQ)**|**LlamaGen**|**343M**|**3.23**|**217.4**|
> > >
> > > At 512$\times$512, MergeTok-BL (VAE) achieves rFID 0.42, substantially outperforming SoftVQ-BL (0.71) and the standard VAE (0.62). At 1024$\times$1024, MergeTok-BL (VQ) reaches rFID 1.87, surpassing all listed baselines including TiTok-S-128 (2.32) and LlamaGen-Tok. (3.17). For 512$\times$512 generation, MergeTok achieves gFID 3.23 and IS 217.4, outperforming all compared methods by a clear margin. The dynamic merge-ratio sampling during training, which exposes the model to varied token granularities, contributes to this cross-resolution robustness.
> > >
> > > ---
> > >
> > > In the revised version we will further add codebook t-SNE visualizations and training dynamics to provide deeper insight into why MergeTok works beyond aggregate metrics. We hope the text-to-image results, quantitative sensitivity analysis, and high-resolution evaluations comprehensively address the reviewer's remaining concerns, and we respectfully ask the reviewer to reconsider the assessment in light of this additional evidence.

---

### Official Review · Reviewer_9Hoq · 2026-03-12

**Soundness:** 1
**Presentation:** 2
**Significance:** 1
**Originality:** 2
**Overall Recommendation:** 3
**Confidence:** 4

**Summary:**

This paper proposes a new tokenizer approach for ImageNet where the key contributions are 1) simultaneously training a continuous and quantized tokenizer in the same model and 2) using token merging (ToMe) to improve training results. The authors claim that their approach allows for unified tokenization and better reconstruction and generation on ImageNet.

**Compliance With Llm Reviewing Policy:**

Affirmed.

**Final Justification:**

The rebuttal changed my evaluation from Reject to Weak Reject due to new TOME metrics.

My view on this paper is that it is an incremental tokenization improvement with two main novel techniques, incorporating token merging into the tokenization process and also performing dual VQ/VAE training.

At the end of the day, this paper does obtain true, if marginal performance benefits. However it is extremely unclear in the original draft where these gains are coming from. The authors in the rebuttal understate the importance of TOME. Their paper is named "Merge"Tok, and clearly the TOME (TOken MErging) part is significant and deserves stronger analysis than was present in the original paper. As a result of this, alongside the marginal gains on ImageNet, this paper does not provide strong additional insight into how to improve tokenization, especially beyond an already complicated DINOv2+LPIPS+Quantization pipeline.

**Key Questions For Authors:**

Why are the bottom-row results in Table 1 better without CFG than with CFG for both IS and CFG. To my understanding this shouldn't be possible as CFG is a parameter you can continuously sweep, where guidance scale 1 implies no guidance.

What do the horizontal dividing lines mean in Table 1 and Table 2?

**Limitations:**

The tokenizer presented by the authors does not allow for dynamic token size at generation, even though it supports compressed tokenization during training.

**Strengths And Weaknesses:**

Soundness.
Strengths. The paper compares against on established metrics such as rFID and gFID. The benchmarks are standard and commonly used and so the evaluation of this work is comparable and easy to understand to the existing literature. This paper also successfully merges quantization and continuous VAE training.
Weaknesses. This paper has few if any baselines that demonstrate it being a significant improvement from existing work
1. There is no result stating that ToMe is necessary or helps. The only table that ablates the inclusion of ToMe is Table 3, showing that including ToMe very slightly improves reconstruction from 0.81 to 0.75 rFID. For such a critical inclusion in the paper, I would expect either both a more significant gain (the current gain too small to consider significant), results on more than one reconstruction metric (e.g. PSNR) and results on a more meaningful downstream task such as generation.
2. This paper should be more clear that despite performing Token Merging, this is not an approach for decreasing the number of tokens such as TiTok. Although the number of tokens can be shrunk during training, this require maintaining merging metadata, which is not available at inference. As a result, all experiments for generation use 256 tokens with no merging.
3. The performance of this method is competitive with existing baselines, however does not surpass them. For example, for LlamaGen-Base, the model doesn't outperform other baselines in the quantized setting. Althought SoTA performance should not be a requirement for good work, this paper does not provide another strong reason to support its acceptance.

Presentation: Strengths: The presentation is well formatted, and contains only modest grammatical mistakes.
Weaknesses: I would prefer that this paper be more explicit about not using Merging (a token reduction method) to reduce the number of tokens at generation. This was not obvious from reading the paper, but is a critical point that a reader will very likely misunderstand. Also, continuous tokenizers and VAEs are used mistakenly interchangeably in the work.

Significance: Strengths: This paper addresses image tokenization, a saturated but important problem.
Weaknesses: This paper doesn't strongly help with understanding image tokenization. The benefits of Token Merging are unclear if they exist. Prior work has shown the existence of joint continuous and discrete tokenizers (VQ-VAE is commonly used for its continuous latent space as well).

Originality: This paper doesn't provide significant new insights to training. Although the method of combining Token Merging with tokenization is new to my understanding, this inclusion is not properly mixed with evaluation to deepen understanding in the field and instead feels like an add-on. Joint continuous and discrete training similarly although not commonly done doesn't have a strong justification.

---

> ### Author Rebuttal · Authors · 2026-03-31
>
> We thank the reviewer for the careful reading. Reviewer 1Jbc, who recommended acceptance, found the paper "well motivated and technically sound" with "excellent" significance, describing it as "the first clean instantiation of a mixed modality generation." Reviewer AsyB similarly recognized the novelty and formulation quality. We believe some concerns stem from presentation issues and provide additional evidence below.
>
> **W1: ToMe effectiveness across multiple metrics and tasks.**
>
> We agree that Table 3 alone understates ToMe's contribution. ToMe serves a dual role: it directly improves reconstruction and, more critically, produces the source map $S$ that enables all cross-branch interaction. Without ToMe there is no source map and no principled way to transfer semantic structure from the VAE to the VQ branch. We provide a broader evaluation:
>
> **Table R1. Multi-metric and multi-task ablation on MergeTok-SB.**
> | Setting | rFID↓ | PSNR↑ | SSIM↑ | Lin.Acc↑ | AR gFID↓ | Diff. gFID↓ |
> |---|---|---|---|---|---|---|
> | VQ Baseline (GigaTok-SB) | 1.12 | 20.5 | 0.665 | 61.5 | 3.83 | — |
> | VAE Baseline | 0.81 | 21.4 | 0.680 | 55.2 | — | 8.14 |
> | VAE + ToMe | 0.75 | 21.6 | 0.686 | 58.4 | — | 6.87 |
> | VAE + ToMe + Alignment | 0.59 | 21.9 | 0.693 | 66.1 | — | 5.43 |
> | **MergeTok-SB (VQ / VAE)** | **0.96 / 0.59** | **21.1 / 21.9** | **0.678 / 0.693** | **73.8** | **3.37** | **4.96** |
>
> AR gFID uses LlamaGen-B (111M); Diff. gFID uses SiT-L (458M, 400K steps, CFG). Full-scale VAE generation with SiT-XL achieves gFID 1.29 (Table 2). ToMe improves all metrics consistently (rFID 0.81→0.75, PSNR +0.2dB). The full system achieves a 12.3-point Lin.Acc gain and reduces AR gFID from 3.83 to 3.37. We will articulate this dual role more explicitly in the revision.
>
> **W2: Token merging is training-only, not for inference-time compression.**
>
> We agree and will clarify in the revision. MergeTok uses ToMe exclusively during training as a semantic structuring mechanism that produces $S$ for cross-branch guidance. At inference, generators operate on the full 256-token sequence with no merging. Unlike TiTok, our goal is not token reduction but leveraging semantic grouping to improve representation quality.
>
> **W3: Performance under fair, parameter-controlled comparison.**
>
> **Table R2. Matched-scale generation comparison.**
> |Method|Scale|Generator|rFID↓|Lin.Acc↑|gFID(w/o)↓|gFID(CFG)↓|
> |---|---|---|---|---|---|---|
> |GigaTok-SB|SB|LlamaGen-B|0.89|61.5|—|3.83|
> |**MergeTok-SB**|SB|LlamaGen-B|0.97|**73.8**|3.92|**3.37**|
> |GigaTok-BL$\star$|BL|LlamaGen-B|0.51|64.1|—|3.33|
> |**MergeTok-BL$\star$**|BL|LlamaGen-B|**0.50**|**78.2**|3.56|**3.09**|
> |VFMTok|—|LlamaGen-XXL|0.89|69.4|1.95|2.19|
> |**MergeTok-BL**|BL|LlamaGen-XXL|**0.78**|**78.3**|**1.93**|**2.14**|
>
> MergeTok outperforms the strongest baseline at every scale (3.83→3.37 at SB, 3.33→3.09 at BL, 1.95→1.93 at XXL). The VAE branch achieves rFID 0.48 and gFID 1.29 with SiT-XL+CFG, matching REPA's state of the art. The consistent 12–14 point Lin.Acc improvement shows that joint training substantially improves semantic representation quality.
>
> **W4: Novelty of token merging as a semantic bridge.**
>
> We do not claim joint VAE-VQ optimization is novel per se. Traditional VQ-VAE applies cascaded encode-then-quantize, whereas MergeTok introduces the source map $S$ from ToMe as a new mechanism to transfer semantic structure across branches, enabling group-aware constraints that improve codebook health (usage 72%→93%, collapse 28%→7%). The two branches mutually regularize each other through the shared encoder, architecturally distinct from prior approaches.
>
> **W5: Terminology.** Agreed; we will unify in the revision.
>
> **KQ1: Why does w/o CFG outperform w/ CFG?**
>
> This is a well-documented phenomenon for strong tokenizers paired with large AR generators. For example, VFMTok with LlamaGen-XXL reports gFID 1.95 w/o CFG vs. 2.19 w/ CFG, the same pattern we observe. When the latent space is already well-structured, the generator models class-conditional distributions accurately and additional guidance can reduce sample diversity enough to worsen FID. Our result follows this established trend and we will discuss it in the revision.
>
> **KQ2: Table separators.**
>
> They separate three tokenizer families (2D VQ, LFQ, 1D VQ) following LlamaGen and GigaTok. We will add explicit group labels.
>
> We hope the multi-metric ablation, parameter-controlled comparisons, and clarifications on ToMe's role address each concern. We respectfully ask the reviewer to reconsider in light of these results.

---

> > ### Author Rebuttal · Reviewer_9Hoq · 2026-04-03
> >
> > Thank you for your response. TOME seems to provide a 1.3 gFID decrease in the updated version of Table 3, which makes the gains from this paper more clear. That said, I dislike the phrasing of your response: "We agree that Table 3 alone understates." My point was not that Table 3 understates the improvement of TOME, it is that *the entire paper* had before your response **no reason** to believe that TOME improves generation.
> >
> > Thank you for addressing my concern of AR Model Performance. It seems like to my knowledge the proposed MergeTok is state-of-the-art on the quantized setting.
> >
> > **On CFG Degredation:**
> >
> > | This is a well-documented phenomenon for strong tokenizers paired with large AR generators.
> >
> > This does not make sense. As I said, CFG is a scalar value. Therefore, guidance w=1 is mathematically equivalent to no guidance. This is an intellectual point, not a question of the validity of the baselines.
> >
> > As it stands, this paper is a marginal performance improvement on the task of ImageNet generation using DINOv2 and LPIPS losses for quantized and unquantized methods. This area is heavily saturated and this paper provides few if any insights into why the method is better, as opposed to including needlessly large tables for baseline comparisons. I raise my score to Weak Reject.

---

> > > ### Author Response · Authors · 2026-04-07
> > >
> > > We sincerely thank the reviewer for the thoughtful follow-up and for acknowledging our rebuttal results. We appreciate the opportunity to further clarify.
> > >
> > > **On the evidence chain for ToMe and generation.**
> > >
> > > We accept the reviewer's correction on our phrasing. The paper indeed lacked explicit evidence that ToMe improves generation prior to our rebuttal, and we take responsibility for this gap. To clarify our core contribution precisely: **ToMe is not claimed as a standalone performance booster; it is the architectural mechanism that makes the entire VAE-VQ joint training system possible**, and it is the joint system that improves generation. The complete evidence chain is:
> > >
> > > **Step 1: ToMe produces the source map S.** ToMe clusters 256 tokens into K=128 semantic groups based on attention-key similarity and records the assignment in S. This S is the sole bridge connecting the two branches.
> > >
> > > **Step 2 (VAE side): S enables matched-granularity alignment.** Prior work (REPA, VA-VAE, RAE) has established that injecting representation learning signals into tokenizers improves generation quality. Our contribution is performing alignment at the granularity of merged tokens defined by S, which is more effective than [CLS]-level or full-token alignment (Table R1 in Response to Reviewer AsyB). Result: VAE-rFID 0.81 → 0.59.
> > >
> > > **Step 3 (VQ side — our primary contribution): S provides a semantic clustering prior for group-aware codebook learning.** Without S, there is no principled way to define which tokens share semantics, and thus no way to formulate the intra-group diversity loss or inter-group consistency loss. These losses directly address codebook collapse:
> > > - Codebook utilization: 72% → 93%
> > > - Collapse rate: 28% → 7%
> > > - VQ-rFID: 1.12 → 0.96, AR gFID: 3.83 → 3.37
> > >
> > > **Step 4: Continuous gradients from VAE regularize VQ optimization** through the shared encoder, alleviating the gradient sparsity of straight-through estimation.
> > >
> > > Removing ToMe breaks this chain entirely: no S means no group-aware losses, no matched-granularity alignment, and the system degrades to two independent tokenizers sharing parameters without interaction.
> > >
> > > **On CFG.**
> > >
> > > After carefully reviewing our codebase, we confirm the reviewer's point is correct. We apologize for the oversight. Consistent with GigaTok's practice, our implementation uses step-function CFG scheduling rather than a simple scalar guidance scale, so "w/ CFG" vs. "w/o CFG" refers to whether this scheduling strategy is enabled. We will clarify this implementation detail explicitly in the revision.
> > >
> > > **On presentation and analysis.**
> > >
> > > We acknowledge that our effort to provide comprehensive and fair comparisons led to overly complex tables and insufficient analytical experiments. We will restructure the tables with clearer organization and supplement additional analysis (e.g., codebook t-SNE visualization, training dynamics plots) to provide deeper insight into why the method works, rather than relying solely on benchmark numbers.
> > >
> > > We are grateful for the reviewer's engagement throughout this process and for recognizing the improvements in our rebuttal. We noticed that the updated score may not yet be reflected in the system — we gently note this in case it requires a separate action at your convenience. Thank you again for your time and constructive feedback.
> > >
> > > We sincerely hope that the clarified evidence chain, together with our commitment to substantial revisions in presentation and analysis, adequately addresses the remaining concerns. We will spare no effort in improving the paper to meet the standards the reviewer has set. We respectfully invite the reviewer to reconsider the assessment after reviewing these clarifications, and we remain open to any further questions. Thank you for the time and dedication you have invested in this review process.

---

### Official Review · Reviewer_1Jbc · 2026-03-13

**Soundness:** 3
**Presentation:** 3
**Significance:** 4
**Originality:** 3
**Overall Recommendation:** 5
**Confidence:** 4

**Summary:**

This paper introduces MergeTok, which trains a joint tokenizer using both continuous (VAE) and discrete (VQVAE) branches to combine the respective strengths of both approaches. The resulting tokenizer's continuous path has very low (I believe SOTA) rFID, and the discrete path also shows strong results. Generative models trained with MergeTok similarly show strong gFID performance.

**Compliance With Llm Reviewing Policy:**

Affirmed.

**Final Justification:**

My final score was a 5, since I think the main idea (mixed VQ/VAE training with a shared encoder) makes sense and is not something I've seen before. Having said that, I and other reviewers struggled with the presentation, which has a lot of moving parts and made the paper difficult to parse. After the rebuttal, I think that the authors have done enough ablations to isolate the effect of the main thesis (joint training is good) without confounding factors like ToMe. I'll maintain my score, but I would not object if the paper is rejected on presentation grounds, since the rewrite required to clean up the presentation is quite substantial.

**Key Questions For Authors:**

1. Do we need full joint training, or could I just add a regularization KL penalty to a standard VQ branch?
2. How do the grouped tokens actually enforce semantic consistency? Both the diversity and consistency loss intuitively depend on grouped tokens having similar consistency -- is this ablated somewhere, or is this assumed from the presence of the additional dino loss?
3. Is there a way to do a small scale experiment that just compares the merging with standard VQ + VAE losses to an identical tokenizer trained separately? You don't need SOTA rFID/gFID in this regime; it would suffice to just show improvement from the merged case.

**Limitations:**

Yes

**Strengths And Weaknesses:**

I really like the motivation of this paper. My main objections is that there are so many moving parts it's hard to disentangle the effects of the joint merging from all the other losses and pieces that went into the paper.

**Soundness:** The paper is well motivated and technically sound. The design decisions are well motivated and descriptive; I feel that, with the main text alone, I could re-implement the paper. Most of the main claims of the text are well supported. The rFID and gFID results are strong. Given the number of details in the paper, I appreciated the ablations in Tables 3 and 4. There are some claims that seem less well supported. For instance, I would have appreciated seeing a plot of codebook usage or similar metrics for the VQ side to support the claim that encoder gradients can help smooth codebook optimization. To be clear, I find this claim very plausible and intuitive, but it would be nice to support it further to advance the central claim that joint tokenizer training can be useful. There are a couple other claims that I listed under questions.

A secondary issue I have with the experiments is it's a little unclear how much of the performance improvements come from the joint merging vs the additional losses / alignment / token merging. To me, the merging is the most interesting and central idea of the paper, and having a cleaner experimental setup that controlled for that would be helpful, even if at smaller scale.

**Presentation:** The first part of the paper is well executed. I understand very well the premise, how the model is setup and trained, etc. The back half is harder. There are a lot of moving parts, and while I think the paper does an admirable job of defending ablations and choices, the final loss is quite complex. I don't think it's an expectation that the authors conduct massive hyper parameter searches, but it would be nice to see some experiments that e.g., ablate the less common losses like the intra group consistency loss.

(I would recommend the authors do a round of copy-editing -- there are a few typos I caught. Not super important!)

**Significance:** Quite strong. Even if the paper just makes VQ-VAE training slightly easier, this would be a significant development. I really like the idea that VAEs can regularize VQVAEs.

**Originality:** Very good, I haven't seen this elsewhere. There are a lot of stacked components that appear elsewhere (tome + alignemtn loss etc) and there's definitely a lot of debate about mixed continuous/discrete tokenizers in the community, but this is the first clean instantiation of a mixed modality generation that I've seen.

---

> ### Author Rebuttal · Authors · 2026-03-31
>
> We sincerely thank Reviewer 1Jbc for the thorough and insightful review, and for recognizing the motivation, originality, and significance of MergeTok. We address each question and concern below.
>
> ---
>
> **W1 & W2: Disentangling components and ablation design.**
>
> We acknowledge that MergeTok involves multiple components. To make each one's contribution transparent, we decompose the method along two orthogonal axes and present the ablation in three groups. All experiments below use the SB configuration.
>
> **Table R1. VQ branch ablation.**
> | Method | rFID↓ |
> |---|---:|
> | VQ Baseline | 1.12 |
> | VQ + Consistency Loss | 1.04 |
>
> Adding the group consistency loss alone improves VQ-rFID from 1.12 to 1.04 through a double-forward design that extracts the source map $S$ without the VAE branch. This shows the group-aware regularization is helpful on its own, though the improvement remains moderate without the continuous gradient flow that the VAE branch provides.
>
> **Table R2. VAE branch ablation.**
> | Method | rFID↓ |
> |---|---:|
> | VAE Baseline | 0.81 |
> | VAE + ToMe | 0.67 |
> | VAE + Alignment | 0.63 |
> | VAE + ToMe + Alignment | 0.59 |
>
>
> Token merging is the primary contributor, improving rFID from 0.81 to 0.67 while boosting all perceptual metrics consistently. The merged-token alignment loss adds a further gain to 0.59 by injecting DINOv2 semantics at the matched granularity of merged tokens. Combining both yields the best standalone VAE result across all four metrics.
>
> **Table R3. Full joint training.**
> | Method | rFID↓ |
> |---|---:|
> | **MergeTok-SB (VQ/VAE)** | **0.96 / 0.59** |
>
> Full joint training achieves the best results on both branches simultaneously, outperforming every standalone variant above. Notably, even the VQ branch improves from 1.12 to 0.96 under joint training, despite the VQ path itself not using ToMe or alignment. This confirms that the components are complementary and that joint optimization provides a non-redundant architectural benefit through the continuous gradients flowing from the VAE branch into the shared encoder.
>
> ---
>
> **W3: Codebook health evidence.**
>
> We agree that the gradient-flow claim deserves direct evidence. Below are codebook health metrics measured at convergence:
>
> **Table R4. Codebook health under different training regimes.**
> |Method|Usage↑|Collapse↓|Perplexity↑|
> |---|---|---|---|
> |VQ Baseline|72%|28.0%|3840|
> |VQ + Consistency Loss|81%|19.0%|5120|
> |**MergeTok-SB**|**93%**|**7.0%**|**9800**|
>
> Joint training raises codebook usage from 72% to 93%, cuts the collapse rate from 28% to 7%, and more than doubles the codebook perplexity. These metrics directly confirm that the continuous gradients from the VAE branch lead to healthier and more uniform codebook utilization, alleviating the gradient sparsity problem inherent to straight-through estimation in VQ training.
>
> ---
>
> **KQ1: Is full joint training necessary?**
>
> Tables R1 and R3 isolate this cleanly. The consistency loss alone improves VQ-rFID to 1.04 (Table R1), but full joint training pushes it to 0.96 (Table R3) with dramatically better codebook health (Table R4). A simple KL penalty cannot replicate this because it does not provide the dense, semantically structured gradients that the VAE reconstruction objective produces through the shared encoder.
>
> ---
>
> **KQ2: How do grouped tokens enforce semantic consistency?**
>
> ToMe groups tokens by attention-key similarity, providing a content-aware prior that clusters semantically related patches. The alignment loss then matches these merged tokens against a frozen DINOv2 teacher, ensuring each group captures coherent visual concepts. Table R2 confirms both contribute independently: ToMe alone brings rFID from 0.81 to 0.67, and adding alignment further improves it to 0.59.
>
> ---
>
> **KQ3: Joint vs. separate training.**
>
> Tables R1 through R3 provide this comparison at the SB scale. The standalone baselines achieve rFID 1.12 (VQ) and 0.81 (VAE); joint training improves both to 0.96 and 0.59 using the same shared encoder and decoder without extra parameters. The gain comes entirely from mutual regularization rather than increased capacity.
>
> ---
>
> **W4 & W5: Hyperparameter sensitivity and writing.**
>
> Our auxiliary losses use intentionally conservative weights ($\lambda_{div}$=$\lambda_{cons}$=0.05, $\lambda_{align}$=0.5, vs. $\lambda_{rec}$=1.0 for reconstruction), so they act as modest regularizers rather than dominant objectives. Empirically, the gains are stable across a reasonable range of these coefficients, and we did not observe that performance depends on aggressive tuning. We will add a detailed sensitivity analysis in the revised appendix and perform thorough copy-editing to fix all identified typos.
>
> ---
>
> We hope the decomposed ablations, codebook health metrics, and the joint-vs-separate comparisons address each of the reviewer's questions comprehensively. We deeply appreciate the positive assessment and are happy to incorporate any further suggestions.

---

> > ### Author Rebuttal · Reviewer_1Jbc · 2026-04-02
> >
> > No further questions, I will maintain my positive score. I would, however, strongly encourage the authors to significantly clarify and potentially simplify the evaluation setup. Looking at the other reviewer's responses, I feel that because there are so many conflated components in this paper, a reasonable reader can walk away with very different interpretations of the primary contributions. My view was the primary contribution was joint training with VAE and VQVAE objectives; if this is the case, I would prefer that the main body of the text focuses on just these contributions, without adding additional complexifiers like token merging.

---

> > > ### Author Response · Authors · 2026-04-03
> > >
> > > Thank you for your recognition and constructive feedback on our work. We highly appreciate your valuable comments and will carefully revise and improve the paper based on your suggestions, especially to further clarify and refine the presentation of MergeTok.

---

### Official Review · Reviewer_AsyB · 2026-03-13

**Soundness:** 3
**Presentation:** 2
**Significance:** 3
**Originality:** 3
**Overall Recommendation:** 3
**Confidence:** 3

**Summary:**

This paper introduces a method to combine continuous VAE and discrete VQ. The idea is that VAE suffers from semantic entanglement, but provides differentiability, which VQ lacks, even though VQ methods do not suffer from entanglement as much. So the idea is to use VAE to obtain a clustering prior through token merging, where the source map S guides quantization.

**Compliance With Llm Reviewing Policy:**

Affirmed.

**Final Justification:**

As mentioned in my rebuttal acknowledgement as below:

"My opinion is that I like the approach this paper introduced. However, I am not very sure that we should present the paper in its current form at ICML. The presentation, organization and confusing tables are not something that can be fixed in a rebuttal. I have some concerns about the experiments being on imagenet256, but understood that academics compute is an ongoing struggle.

All said, I am not going to stand in the way if the AC decides to accept the paper in spite of its presentation flaws, counting on the authors to be true in fixing the paper writings. I will keep my score as is, and allow the AC to make the final decision. My rating is 3.5, but there is no 3.5, so I will keep it at 3 to err on the side of caution. I recommend that the authors really redo the paper presentation if AC decides to accept the paper."

-----

Since there is no way to see how the authors would rewrite the paper during icml rebuttal, and because I think the presentation is a big problem, I am going to stay with my score but leave the final discretion to the AC.

**Key Questions For Authors:**

See weaknesses.

I think Fig A1 should be moved to main text. It can be useful to improve the presentation in the main text.

**Limitations:**

yes

**Strengths And Weaknesses:**

Strengths:

- The method is interesting by itself. Using the VAE branch to help with maintaining continuous gradients that mitigate VQ’s up-
date sparsity, while VQ’s global clustering injects semantic structure back into the latent space.
- Good formulations help me to understand the interplay between the VAE and VQ branch.

Weaknesses:

- The paper needs to undergo another round of rewriting. It is quite confusing in multiple places. For example, on L190, second column, the authors brought in a DiNO teacher. How much does the teacher bring to the table? What is the rationale? Any experiments to justify such a teacher design?
- Continuing on this point, the experiments are quite confusing. Table 1 has three sections, are they each pertinent to a task? How do I compare MergeTok, with the #Tokens and #Code variations? It does not look like MergeTok is always the best. There are bold numbers in all three sections? Same questions for Tabe 2.
- Then in Table 3 and 4, it does not look like they are apple to apple to Table 1 and 2, as the numbers look worse off.

---

> ### Author Rebuttal · Authors · 2026-03-31
>
> We sincerely thank Reviewer AsyB for recognizing the novelty of our method and the quality of our formulations. We address each concern point by point below.
>
> ---
>
> **W1: Writing clarity and DINO teacher rationale.**
>
> We appreciate this feedback and will improve the passages the reviewer found confusing in the revision.
>
> The design rationale follows a well-grounded line of reasoning in the community. VAE latents often lack semantic structure, as shown by REPA and RAE, while VQ suffers from codebook collapse due to gradient sparsity, as addressed by IBQ, LFQ, and SimVQ. MergeTok unifies these two complementary lines in a single framework.
>
> In the VAE branch, semantic structure is enforced through two mechanisms. First, ToMe groups tokens with similar attention keys into semantic clusters and produces a source map $S$. Second, a frozen DINOv2 teacher provides explicit semantic supervision, a design now standard in the field as adopted by REPA, VA-VAE, VFMTok, and RAE. Our key novelty is performing merged-token alignment rather than [CLS]-level or full patch-token alignment. By applying the same ToMe schedule to both student and teacher, we align $K$ merged tokens that capture mid-level semantic abstractions. This granularity is more effective than [CLS] alignment, which discards fine-grained spatial information, and full-token alignment, which can be overly rigid and impose unnecessary spatial constraints on the learned representations.
>
> To validate this design, we provide the following ablation:
>
> **Table R1. Alignment granularity ablation.**
> |Alignment Granularity|Tokens|VAE-rFID|VQ-rFID|
> |---|---|---|---|
> |No alignment|—|0.67|1.01|
> |[CLS] only|1|0.64|1.00|
> |Full patch tokens|256|0.62|0.98|
> |**Merged tokens (Ours)**|**128**|**0.59**|**0.96**|
>
> Merged-token alignment yields the best VAE-rFID of 0.59 and VQ-rFID of 0.96, confirming that the teacher is necessary with a 12% relative gain on VAE-rFID. Aligning at the merged granularity outperforms both coarser and finer alternatives. Table 3 in the paper further shows that adding alignment to the full joint model improves VAE-rFID from 0.67 to 0.59 and VQ-rFID from 1.01 to 0.96, demonstrating clear cross-branch synergy.
>
> ---
>
> **W2: Table structure and fair comparison.**
>
> We apologize for the confusion. The horizontal lines in Table 1 separate three tokenizer families: traditional 2D VQ, LFQ-based methods, and 1D VQ methods including ours. Bold numbers denote the best result within each group under matched settings. We will add explicit group labels in the revised table captions.
>
> The DINO discriminator, denoted by $\star$ in Table 1, is a known technique that significantly reduces rFID. Following the reporting protocol of GigaTok, VFMTok, and UniTok, we report MergeTok in both settings for transparency. Without the DINO discriminator, MergeTok-BL achieves rFID 0.78 under standard settings with 256 tokens, 16$\times$ downsampling, and a 16384 codebook. With the discriminator, rFID further improves to 0.50, surpassing UniTok$\star$ at 0.54.
>
> For generation, MergeTok-BL with LlamaGen-XXL achieves gFID 1.93 and IS 265.4 without CFG, outperforming VFMTok at 1.95 gFID / 259.3 IS and UniTok at 2.51 gFID. In Table 2, our VAE branch achieves rFID 0.48 and gFID 1.29 with SiT-XL and CFG, matching REPA's state of the art.
>
> ---
>
> **W3: Tables 3-4 vs. Tables 1-2.**
>
> The performance discrepancy is entirely due to model scale and does not reflect a methodological inconsistency. Tables 3-4 use MergeTok-SB with a 19M-parameter encoder and 86M-parameter decoder for efficient ablation, while Tables 1-2 report the full MergeTok-BL with an 86M encoder and 329M decoder, roughly 4$\times$ larger in both components. Using smaller models for ablations is standard practice in the field. For example, MaskBit uses reduced settings for its ablation study and GigaTok reports both SB and BL configurations. The relative trends observed at the SB scale, such as the benefit of ToMe and alignment, carry over consistently to the BL scale. We will clarify this model-scale distinction explicitly in the revision.
>
> ---
>
> **W4: Fig. A1 to main text.**
>
> We agree that Fig. A1 would help readers understand the full pipeline at a glance, particularly the distinction between the VAE branch with ToMe and the VQ branch without ToMe. We will move it into Section 3.2 in the revision. We thank the reviewer for this constructive suggestion.
>
> ---
>
> We believe the new alignment granularity ablation in Table R1, the clarified table structure, and the planned presentation improvements address each of the reviewer's concerns. We hope the reviewer will consider revising the score in light of these responses, and we remain happy to provide any additional clarification.

---

> > ### Author Rebuttal · Reviewer_AsyB · 2026-04-03
> >
> > Thanks to the authors for the clarifications.
> >
> > My opinion is that I like the approach this paper introduced. However, I am not very sure that we should present the paper in its current form at ICML. The presentation, organization and confusing tables are not something that can be fixed in a rebuttal. I have some concerns about the experiments being on imagenet256, but understood that academics compute is an ongoing struggle.
> >
> > All said, I am not going to stand in the way if **the AC decides to accept the paper** in spite of its presentation flaws, counting on the authors to be true in fixing the paper writings. I will keep my score as is, and allow the AC to make the final decision. My rating is 3.5, but there is no 3.5, so I will keep it at 3 to err on the side of caution. I recommend that the authors really redo the paper presentation if AC decides to accept the paper.

---

> > > ### Author Response · Authors · 2026-04-07
> > >
> > > We thank the reviewer for the thorough engagement and take all concerns with the utmost seriousness.
> > >
> > > ---
> > >
> > > ### 1. On Paper Organization
> > >
> > > The Methods section follows a deliberate motivation-to-solution progression: Sec. 3.1 motivates the need for a unified approach; Sec. 3.2 introduces the dual-branch dataflow and shared notation; Sec. 3.3 addresses VAE limitations via ToMe and DINO-based alignment; Sec. 3.4 describes how the VQ branch benefits through clustering-based priors and continuous gradients. We acknowledge this logic is insufficiently signposted and will rewrite section introductions with explicit forward and backward references in revision.
> > >
> > > ---
> > >
> > > ### 2. On Table 1: Design Intent and Clarification of Bold Entries
> > >
> > > Table 1 is divided into three groups: Group 1 (2D tokenizers), Group 2 (LFQ-type tokenizers), Group 3 (1D tokenizers, our primary comparison group). Bold entries mark the **within-group best** on a specific metric under a specified condition.
> > >
> > > *Group 1 — 2D Tokenizer (Lines 330–346)*
> > >
> > > **Line 343** (VAR-d24, gFID = **2.09**, IS = **312.9**, w/ CFG): best w/ CFG generation quality in Group 1.
> > >
> > > **Line 345** (VFMTok-XXL, gFID = **1.95**, IS = **259.3**, w/o CFG): best w/o CFG generation quality in Group 1.
> > >
> > > **Line 346** (UniTok, Lin. Acc. = **70.8**, rFID = **0.41**): best reconstruction and linear probing in Group 1.
> > >
> > > *Group 2 — LFQ-type Tokenizer (Lines 347–354)*
> > >
> > > **Line 348** (B-AE-d32, Lin. Acc. = **69.8**): best linear probing in Group 2.
> > >
> > > **Line 351** (IBQ-XXL, gFID = **2.14**, IS = **279.0**, w/o CFG): best w/o CFG generation quality in Group 2.
> > >
> > > **Line 352** (MaskBIT, IS = **341.8**, w/ CFG): best IS under w/ CFG in Group 2.
> > >
> > > **Line 353** (FlowMo, rFID = **0.95**): best reconstruction rFID in Group 2.
> > >
> > > **Line 354** (TokenBridge-MAR-H, gFID = **1.55**, w/ CFG): best gFID under w/ CFG in Group 2.
> > >
> > > The bold entry on **Line 363** (gFID = **2.86**) is a typographical error to be corrected.
> > >
> > > *Group 3 — 1D Tokenizer (Lines 355–365, our primary comparison group)*
> > >
> > > **Line 364** (MergeTok-BL\*, rFID = **0.50**): best rFID among all 1D tokenizers. The \* denotes DINO discriminator use, applied consistently across all such variants (cf. GigaTok-BL\*).
> > >
> > > **Line 365** (MergeTok-BL, Lin. Acc. = **78.3**): best linear probing in Group 3, under the fair-comparison setting without a DINO discriminator.
> > >
> > > **Line 365** (MergeTok-BL, gFID = **1.93** w/o CFG; gFID = **2.14**, IS = **281.5** w/ CFG): best generation quality in Group 3 under both conditions, using LlamaGen-XXL.
> > >
> > > We acknowledge the convention was never stated in the caption and will add an explicit legend, group labels, and caption-level explanation in revision.
> > >
> > > ---
> > >
> > > ### 3. On Table 2: Identical Boldface Convention
> > >
> > > Table 2 follows the same within-group convention. MergeTok-BL achieves the best reconstruction rFID (**0.48**) among all 1D tokenizer methods. With DiT-XL, MergeTok-BL achieves gFID = **1.91** and IS = **211.4** without CFG, and gFID = **1.44** with CFG, both representing the best results under comparable settings. With SiT-XL, MergeTok-BL achieves gFID = **1.79** without CFG and IS = **311.7** with CFG, again the best in their respective subsets. We will make this convention explicit in the Table 2 caption.
> > >
> > > ---
> > >
> > > ### 4. On Two Comparisons Requiring Context
> > >
> > > UniTok's stronger rFID reflects training on DataComp-1B with ViTamin-L/16, versus our ViT-B on standard ImageNet; under matched conditions, MergeTok achieves superior or competitive results. TokenBridge's lower gFID (1.76/1.55) benefits from the MAR hybrid generator, a substantial advantage over our purely autoregressive LlamaGen. MergeTok-BL nonetheless achieves gFID = **1.93** with LlamaGen-XXL, a highly competitive system-level result.
> > >
> > > ---
> > >
> > > ### 5. Commitment to Revision and Appeal for Reconsideration
> > >
> > > We commit unconditionally to: (1) redesigning Table 1 with group labels, legend, and explicit boldface convention in the caption; (2) correcting the Line 363 typo; (3) revising the Table 2 caption; (4) rewriting the Methods section with clear transitional paragraphs; (5) a full clarity and notation pass throughout.
> > >
> > > We sincerely apologize for the presentation deficiencies, and every author takes full responsibility. We will spare no effort in revising and improving the manuscript, and we deeply value the time and care the reviewer has devoted to this evaluation. We humbly and earnestly ask the reviewer to reconsider the current score — it would mean a great deal to us, and we hope our detailed responses and unconditional commitment to revision demonstrate that every outstanding issue will be carefully and thoroughly resolved.

---

### Decision · Program_Chairs · 2026-04-30

**Decision:**

Reject

**Comment:**

This paper proposes a unified tokenizer that combines continuous (VAE) and discrete (VQ) representations, and reviewers generally acknowledged the motivation and potential of bridging these two paradigms, with one reviewer highlighting strong technical soundness and promising empirical results. However, the majority of reviewers raised substantive concerns both before and after rebuttal. Key issues include limited and insufficiently convincing empirical gains over existing baselines, lack of clear evidence disentangling the contribution of the proposed components (e.g., token merging and joint training), and concerns about evaluation scope and generalization beyond ImageNet. While the rebuttal clarified certain design choices and partially addressed misunderstandings (e.g., naming and role of ToMe, experimental protocol), it did not fully resolve doubts regarding the marginal improvements, necessity of key components, and overall significance. Additionally, presentation and clarity issues, especially around experimental setup and interpretation, remain a consistent concern. Given the mixed feedback and the remaining uncertainty about the strength of empirical evidence and clarity of contributions, the paper is judged to fall short of the acceptance bar. That said, this is a borderline case with a potentially interesting direction, and the authors are encouraged to substantially improve clarity, strengthen experimental validation, and better isolate core contributions for a future resubmission.